# An empirical energy landscape reveals mechanism of proteasome in polypeptide translocation

**Rui Fang, Jason Hon, Mengying Zhou, Ying Lu\***

Department of Systems Biology, Harvard Medical School, Boston, United States

**Abstract** The ring-like ATPase complexes in the AAA+ family perform diverse cellular functions that require coordination between the conformational transitions of their individual ATPase subunits (Erzberger and Berger, 2006; Puchades et al., 2020). How the energy from ATP hydrolysis is captured to perform mechanical work by these coordinated movements is unknown. In this study, we developed a novel approach for delineating the nucleotide-dependent free-energy landscape (FEL) of the proteasome's heterohexameric ATPase complex based on complementary structural and kinetic measurements. We used the FEL to simulate the dynamics of the proteasome and quantitatively evaluated the predicted structural and kinetic properties. The FEL model predictions are consistent with a wide range of experimental observations in this and previous studies and suggested novel mechanistic features of the proteasomal ATPases. We find that the cooperative movements of the ATPase subunits result from the design of the ATPase hexamer entailing a unique free-energy minimum for each nucleotide-binding status. ATP hydrolysis dictates the direction of substrate translocation by triggering an energy-dissipating conformational transition of the ATPase complex.

## Editor's evaluation

The present work is important for using innovative computational approaches and biochemical analyses to help to explain how hexameric peptide translocases and unfoldases belonging to AAA+ ATPases couple nucleotide turnover to directed chain movement. The work sheds light on understanding not only normal, processive translocation but also how the motors can operate with a defective subunit.

**\*For correspondence:**
ying_lu@hms.harvard.edu

**Competing interest:** The authors declare that no competing interests exist.

## Introduction

The ring-shaped oligomeric ATPases control key biological processes including protein folding, transcription, DNA replication, cellular cargo transport, and protein turnover (*Erzberger and Berger, 2006*; *Puchades et al., 2020*; *White and Lauring, 2007*). The 26S proteasome is an ATP-dependent protein degradation machine in the AAA+ (ATPases associated with diverse cellular activities) family of ATPases (*Collins and Goldberg, 2017*). The proteasome holoenzyme consists of a barrel-shaped 20S core particle (CP) capped by 19S regulatory particles (RPs) on one or both ends (*Bard et al., 2018*). Each RP features a nine-member Lid subcomplex and a heterohexameric ring of AAA+ ATPases assembled from six distinct gene products (RPT1–RPT6) that share 85% sequence identity. These ATPases use the energy from ATP hydrolysis to mechanically unfold substrates and translocate them into the CP for proteolysis (*Figure 1A*).

Processive translocation and degradation of protein substrates is critical for the biological functions of the ubiquitin-proteasome system (*Collins and Goldberg, 2017*; *Hershko and Ciechanover, 1998*). In previous investigations of the structural mechanism of proteasomal degradation, cryo-electron

**eLife digest** In cells, many biological processes are carried out by large complexes made up of different proteins. These macromolecules act like miniature machines, flexing and moving their various parts to perform their cellular roles. One such complex is the 26S proteasome, which is responsible for recycling other proteins in the cell. The proteasome consists of approximately 31 subunits, including a ring of six ATPase enzymes that provide the complex with the energy it needs to mechanically unfold proteins.

To understand how the proteasome and other large complexes work, researchers need to be able to monitor how their structure changes over time. These dynamics are challenging to probe directly with experiments, but can be assessed using computer simulations which track the movement of individual molecules and atoms. However, currently available computer systems do not have enough power to simulate the dynamics of large protein assemblies, like the 26S proteasome: for example, it would take longer than a thousand years to model how each atom in the complex moves over a timescale in which a biological change would happen (roughly 100ms).

Here, Fang, Hon et al. have developed a new approach to simulate the structural dynamics of the proteasome's ring of ATPase enzymes. Different known structures of the proteasome were used to identify the range of possible movements and shapes the complex can make. Fang, Hon et al. then used this data to calculate the energy level of each structure – also known as the 'free energy landscape' – and the rate of transition between them. This made it possible to simulate how the different ATPase enzymes move within the ring under a wide range of conditions.

The simulated ATPase movements predicted how the proteasome machine would behave during various tasks, including degrading other proteins. Fan, Hon et al. carefully examined these predictions and found that they were consistent with experimental observations, validating their new simulation method.

This work demonstrates the feasibility of simulating the actions of a large protein complex based on its free energy landscape. The results offer important insights into the functional mechanics of the 26S proteasome and related protein machines. Further work may help to simplify this process so the approach can be used to investigate the dynamics of other protein assemblies.

microscopy (cryo-EM) of the substrate-engaged proteasome captured seven states, $E_{A1}$–$E_{D2}$. $E_{A1}$ and $E_{A2}$ likely represent the resting states of proteasome; $E_{D1}$ and $E_{D2}$ are hypothesized to be involved in substrate translocation (*Figure 1—figure supplement 1*; *Dong et al., 2019*; *de la Peña et al., 2018*). In these structures, an unfolded substrate primarily interacts with aromatic residues on the pore-1 loop (PL1) of each ATPase. These short structured loops form a right-handed helical 'staircase' delineating the interior of the translocation channel. Changes in bound nucleotides at the ATPase interfaces are associated with rearrangements of the architecture of the PL1 staircase and the proteasome-substrate interaction, as a result of both translational and pivot movements of these ATPases (*Figure 1B* and *Figure 1—figure supplement 1*; *Dong et al., 2019*; *de la Peña et al., 2018*). One or two PL1s are disengaged from substrate interaction in each state. These disengaged PL1s usually occupy distal, or top, positions in the staircase away from the CP (*Figure 1A*). To account for substrate translocation, we and others have proposed that when certain PL1s disengage and move to the top, substrate-engaged PL1s may move in the opposite direction, toward the CP. This conformational rearrangement may provide the power stroke to promote axial translocation of substrate (*Figure 1B*; *Dong et al., 2019*; *de la Peña et al., 2018*).

In spite of these structural insights, important questions regarding the dynamics of proteasomal structures remain, in particular how the six proteasomal ATPases coordinate during conformational transitions to achieve substrate translocation. Based on these cryo-EM structures, we previously proposed that the conformation and nucleotide-binding status of the ATPase hexamer may cycle consecutively through six different states with rotational equivalence, thus driving processive substrate translocation (*Dong et al., 2019*; *Hua et al., 2002*). A similar model of sequential transitions was proposed in an independent study of yeast proteasome structures (*de la Peña et al., 2018*). Only two out of the six conformations proposed in the sequential-transition model were identified in these structural studies, and direct experimental study of the transition sequence has not been reported

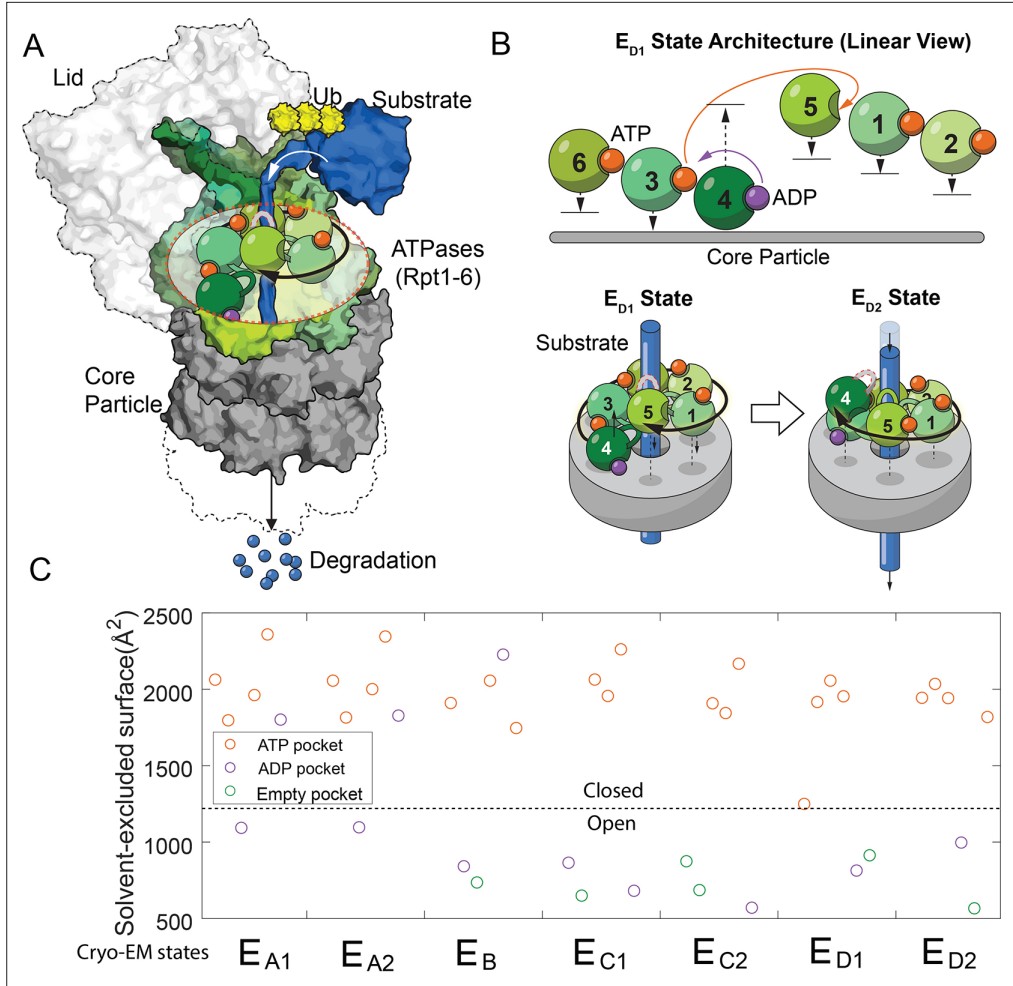

**Figure 1.** The architectures of the proteasomal ATPase complex and its interaction with substrate. (**A**) A schematic showing a half 26S proteasome engaged with an unfolded substrate through the PL1s (color loops) on the ATPase subunits with bound nucleotides (color blobs). The disengaged PL1 is marked in gray. A black arrow suggests the staircase architecture of the ATPases. (**B**) Upper panel: a linear view of the architecture of the ATPase complex in a translocating state $E_{D1}$. Each ATPase subunit (Rpt1–Rpt6) is shifted vertically according to the position of its PL1 relative to the core particle. An open interface is suggested by a large gap between subunits. Arrows indicate the displacements of the PL1s and the change of nucleotides in the $E_{D1}$-to-$E_{D2}$ transition. See *Figure 1—figure supplement 1* for a complete list of these identified states of the proteasome. The lower panel illustrates how the ATPase rearrangement in the $E_{D1}$-to-$E_{D2}$ transition may drive substrate translocation: the PL1 on Rpt4 disengages from substrate and moves to the distal registry of the staircase. This change is accompanied by an axial movement of the PL1s on Rpt1/2/6/3 that still interact with the substrate toward the core particle to bring about axial stepping and translocation of approximate 2× AAs. (**C**) The solvent-excluded surface area at the interfaces of the ATPase domains in the seven substrate-engaged proteasome structures, colored according to the nucleotide at each binding pocket. The dashed line separates the closed and open interfaces, as defined here.

The online version of this article includes the following figure supplement(s) for figure 1:

**Figure supplement 1.** Schematics showing the key structural features and the nucleotide-binding status of the substrate-engaged proteasome identified in a cryo-EM study.

**Figure supplement 2.** Comparison between open and closed interfaces of ATPases on proteasome.

(***Dong et al., 2019***; *de la Peña et al., 2018*). Transiting into the next state involves multiple changes in nucleotide status and a complex conformational rearrangement of the ATPase hexamer. We still do not know the order of these events and how the chemical energy from ATP hydrolysis may be harvested to drive these transitions.

In addition, there are experimental findings that appear inconsistent with the predictions of a strict sequential-transition model. For example, mutations of the Walker-A (WA) or Walker-B (WB) motifs on an ATPase impede its nucleotide-binding or hydrolysis activity, and are therefore predicted to inactivate the proteasome by blocking the transition sequence. However, Walker mutations on some ATPases of the proteasome are in fact well-tolerated in yeast (*Rubin et al., 1998*; *Eisele et al., 2018*; *Kim et al., 2013*). Similarly, mutations of other functional motifs on different proteasomal ATPases have varying effects on protein degradation, leading to the hypothesis that the six ATPases may have nonequivalent roles in the proteasome activities despite their high levels of similarity in sequence and structure (*Rubin et al., 1998*; *Beckwith et al., 2013*; *Erales et al., 2012*). These functional properties of the proteasome are not interpreted by the previous models and so far no alternative model that is consistent with both structural and functional observations has been suggested.

Computational approaches, such as molecular dynamics simulations, are frequently employed to decipher the structural dynamics of proteins (*Brini et al., 2020*). However, despite recent advances, it is still impractical to perform a full-atom simulation of a large system such as the proteasome at a biologically relevant time scale (~100 ms) (*Gecht et al., 2020*). We therefore developed a novel approach to simulate the conformational dynamics of the proteasomal ATPase hexamer by constructing a physical model based on the nucleotide-dependent free-energy landscape (FEL) of the ATPase complex. To obtain the FEL, we first performed comparative analysis on known proteasome structures to identify the primary degrees of freedom (DOFs) of proteasome's conformational changes, and applied these DOFs as the conformational coordinates of the FEL. We then parameterized the free energy surface based on the mode of nucleotide-ATPase interactions, and experimentally determined the nine parameters of the FEL. Conformational changes of the ATPase complex are mapped to simple stochastic transitions on the FEL which evolves as the nucleotide status changes, driven by independent chemical processes at each nucleotide-binding pocket.

To address whether the simulated dynamics based on the FEL model recapitulates the actual dynamics of the ATPase complex, we experimentally tested the model predictions in a wide range of conditions that are distinct from the results used for model construction, as well as comparing the predictions with published results. We found that the FEL predictions were widely congruent with the experimental measurements in this and previous studies.

Our work introduces a new method for studying the dynamics of a complex protein machine such as the 26S proteasome, and provides a coherent explanation for a variety of structural and kinetic observations. It also provides a satisfying picture of the underlying mechanism by which the AAA+ hexamer on the proteasome operates in driving substrate translocation.

## Results

### Structure-based construction of the FEL of the proteasomal ATPase complex

In this work, we elect a combination of the ATPase complex's conformation and the nucleotide distribution in the six ATPase pockets to specify a 'state' of the proteasome, for consistency with nomenclature in structural studies. A description of the FEL of proteasome is represented by the potential of mean force of a specific proteasome population measured as a bivariate function of its conformational coordinates and the nucleotide distribution (*Kirkwood, 1935*; *Rodnina et al., 2011*).

The conformational coordinates of the FEL are defined by the primary DOFs of proteasome's conformational changes, identified by comparing proteasome structures. We designate each conformation in the FEL by whether the interfaces of the six ATPase domains on proteasome are open or closed, based on the observation that the solvent-excluded surface area (SESA) of these interfaces mostly adopts binary values in these cryo-EM structures (*Figure 1C* and *Figure 1—figure supplement 2*). A large SESA is associated with closed interfaces that appear to arrange the PL1s on neighboring ATPases into a staircase that interacts with substrate peptide. In contrast, a smaller SESA suggests an open interface and the relative geometry of the neighboring ATPases tends to vary (*Figure 1—figure supplement 1*). An ATPase subunit that is flanked by two open interfaces is disengaged from substrate interaction, as observed in the cryo-EM structures (*Figure 1—figure supplement 1*). In the FEL, we constrain the total number of open interfaces in each hexamer conformation to be either 2 or 3, as observed in the cryo-EM structures except for the $E_A$ states which involve only one open interface

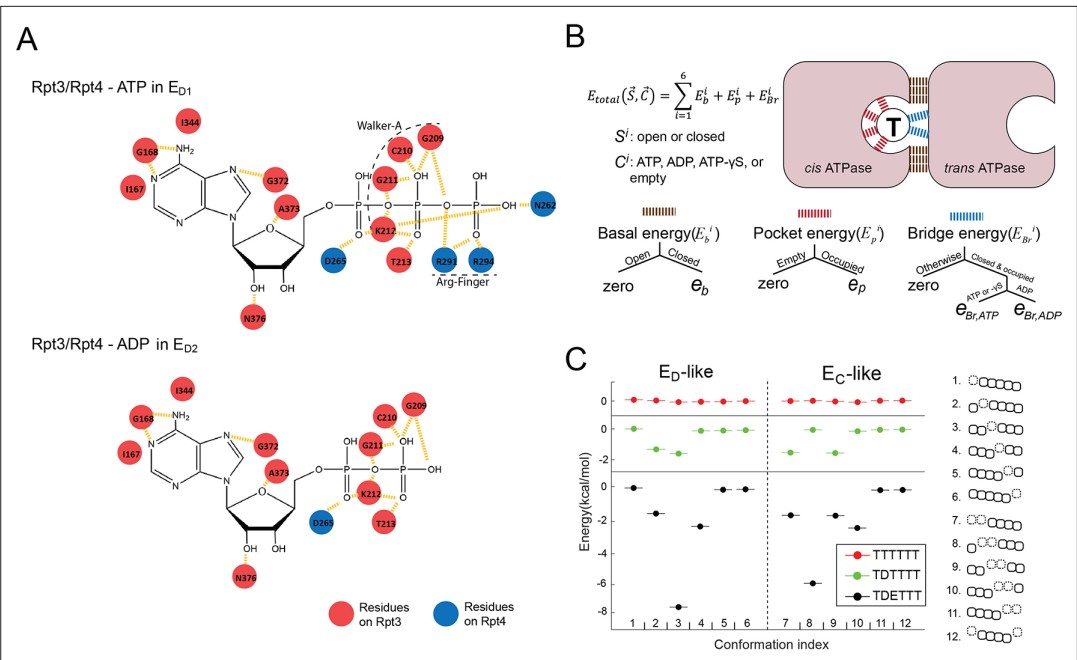

**Figure 2.** Parameterization of the nucleotide-dependent free-energy landscape. (**A**) The interaction map of the residues on Rpt4 and Rpt3 with the bound ATP or ADP in the $E_{D1}$ or $E_{D2}$ states. Red: *cis*-interacting residues on Rpt3. Blue: *trans*-interacting residues on Rpt4. (**B**) A schematic showing the parameterization strategy for the free energy ($E_{total}$) of the ATPase hexamer and valuation of the parameters. The molecular interactions underpinning the three energy terms are marked by different colors. 'Open/Closed' refers to the status of an ATPase interface $S^i$. 'Empty/Occupied/ADP/ATP/ATP-γS' refers to the status of the nucleotide-binding pocket in the *cis* ATPase $C^i$. See 'Determining the FEL and kinetic parameters' for a detailed explanation. (**C**) The FEL on the $E_D$-like and $E_C$-like conformations of the ATPase complex in three representative nucleotide-binding statuses. T/D/E: ATP/ADP/empty. A sketch for the ATPase architecture of each conformation is listed on the right. The ATPases Rpt6/3/4/5/1/2 are represented by squares from left to right; dashed squares=disengaged ATPases that are flanked by open interfaces (gap between squares); the axial position of a PL1 is indicated by the vertical shift of the corresponding ATPase square.

The online version of this article includes the following figure supplement(s) for figure 2:

**Figure supplement 1.** Phosphate groups interact weakly with the proteasomal ATPases.

**Figure supplement 2.** Root-mean-square deviation (RMSD) of the nucleotide-interacting residues in the *cis* pockets among different proteasomal states.

(*Figure 1—figure supplement 1*) (see Discussion). This defines 30 conformations, after excluding 5 with ambiguity in assigning ATPase-substrate interaction due to symmetry (*Supplementary file 1*; Materials and methods 'Defining a discrete conformational space of the proteasome ATPase complex by extrapolating the cryo-EM observations'). States in the FEL are also differentiated by the nucleotide distribution in these ATPases. Each nucleotide pocket may be occupied by ATP, ADP, ATP-γS (a slowly-hydrolyzing ATP analog), or no nucleotide. We ignore the transient ADP•Pi state due to the very weak affinity between free phosphate and the proteasome (*Figure 2—figure supplement 1*); also, no such state has been identified so far in proteasome structures. The total number of distinct states in the FEL model is therefore $30 \times 4^6 = 122,880$.

The molecular details of the nucleotide-ATPase interactions suggest a strategy to parameterize the free energy of each conformation as a function of its nucleotide status. Each nucleotide-ATPase interaction on the proteasome involves several conserved elements: the WA, WB, sensor I, and sensor II motifs from the *cis* ATPase and the arginine fingers from the *trans* ATPase (*Figure 2A*; *Ogura and Wilkinson, 2001*). The arginine fingers are the major components interacting with the γ- and β-phosphate groups on ATP at a closed interface. We found that the *cis* elements exhibited rather minor rearrangements among different states of either substrate-free or substrate-engaged proteasomes (root-mean-square deviation [RMSD] ~0.58 Å) (*Figure 2—figure supplement 2*). We therefore

parameterized the total free energy of the ATPase hexamer as a sum of the contributions from each individual ATPase interface. Each interface's contribution is subdivided into three components: the basal energy $E_b$, which derives from direct interactions between adjacent ATPases at a closed interface; the pocket energy $E_p$ from the nucleotide-*cis*-element interactions, which are similar for ADP and ATP; and the bridge energy $E_{Br}$, which differentiates ADP from ATP and originates from the engagement of arginine fingers and other *trans* elements with the γ- and β-phosphate on the nucleotide at a closed interface (*Figure 2B*). The *cis* pockets of disengaged ATPases exhibit either low or no nucleotide density in cryo-EM maps, and are associated with slightly rearranged *cis* motifs ($p$=0.014) (*Figure 1—figure supplement 1* and *Figure 2—figure supplement 2*; *Dong et al., 2019*). We therefore assign a separate pocket energy $E_p^{APO}$ for disengaged ATPases to reflect their low affinity for nucleotides. The chemical energy in ATP is excluded from the free energy calculation, since it does not explicitly contribute to the simulation of ATPase dynamics (see the next section).

These energy parameters can in principle vary for the different ATPase subunits Rpt1–Rpt6. The magnitude of this variation is unclear. To reduce the model's complexity, we made an approximation that all six ATPases share an identical set of parameters. As described below, this approximation provides excellent agreement with kinetic measurements and is compatible with the asymmetric cryo-EM occupancies and effects of WB mutations. In a following section , we analyze the contribution of the proteasome Lid-ATPase interaction to the symmetry breaking among these ATPases.

## Evaluating the FEL model parameters

We determined all nine parameters in the FEL model from the measurements in this study and previous work (see Materials and methods 'Determining the FEL and kinetic parameters' and 'Determining the parameters related with substrate translocation'). A critical parameter is the difference in the bridge energies ($E_{Br}$) between ATP and ADP interfaces, which we find is key for determining the directionality of translocation. This parameter was measured using a single-molecule binding assay, based on the relationship between the dissociation constant ($K_d$) of the nucleotides and these energy terms. In the FEL framework, the nucleotide pockets are classified into three groups, according to how the $K_d$ depends on the energy parameters (*Figure 3A*). This designation varies with the status of the *cis* ATPase and the interface. We used a very low concentration (200 nM) of Alexa647-conjugated ATP to limit the interaction to the strongest binding pockets (group 1 in *Figure 3A*) on the human 26S proteasome which was immobilized on a passivated coverslip, and observed the interaction between ATP and the proteasome by TIRF (Total Internal Reflection Fluorescence) microscopy. The proteasome was fluorescently labeled on the 19S particle using a SNAP tag to allow accurate detection of colocalization with Alexa647-ATP. We varied the concentrations of competing unlabeled nucleotides (ATP, ADP, and ATP-γS) in a steady-state measurement, to circumvent the possibility that conjugation with fluorophore may alter the nucleotide-ATPase affinity (*Figure 3B*). ATP-γS was employed to minimize nucleotide hydrolysis. This analysis yielded the following ratio of the inhibitor constant ($K_i$) for ATP:ADP:ATP-γS=1:7.9 (±2.0):0.5 (±0.15) (at 90% confidence interval), giving $e_{Br, ATP}-e_{Br, ADP}$=−1.6 (±0.4) kcal/mol.

After determining the FEL parameters, we explored the basic features of the landscape. For the 12 ATPase-hexamer conformations that are characterized by either a single or two adjacent disengaged ATPase units, as in the observed $E_D$ or $E_C$ states, uniform ATP binding generates a flat FEL. The presence of ADP in a single pocket breaks the symmetry and lowers the relative free energy of a subset of conformations by 1.6 kcal/mol. The further addition of an empty pocket next to the ADP pocket produces a well-separated energy-minimum conformation in which the empty and ADP *cis* pockets are found in the disengaged ATPase and its counterclockwise neighbor (*Figure 2C*). Although we did not impose any coupling between nucleotide status and ATPase conformation, this energy-minimum arrangement of nucleotide and conformation is identical to those observed in the $E_{D1}$ and $E_{D2}$ states of proteasome, possibly explaining the prevalence of this arrangement in proteasome structures as well as in related AAA+ ATPases (*Figure 1—figure supplement 1*; *Dong et al., 2019*; *Majumder et al., 2019*; *Cooney et al., 2019*).

We simulated the dynamics of the ATPase complex as stochastic transitions in a discrete 12 (6 conformational+6 nucleotide) dimensional space (*Figure 3—figure supplement 1*). The transitions between conformations were described as a simple bi-state process with a rate constant determined by the Arrhenius equation. The ATP cycle at each pocket proceeds independently, as described by

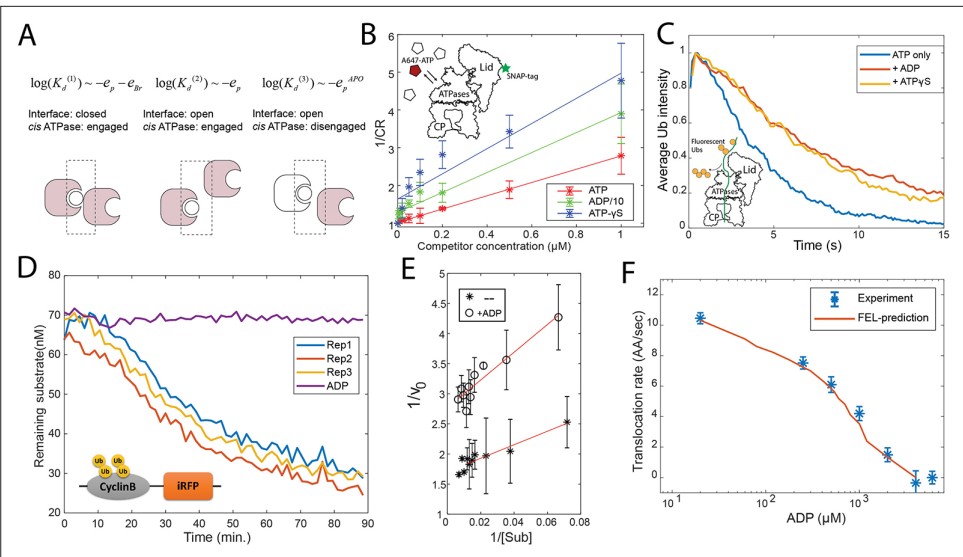

**Figure 3.** Evaluation of the FEL model parameters. (**A**) A schematic showing three categories of nucleotide pockets with their corresponding dissociation constants $K_d$. (**B**) Single-molecule nucleotide-proteasome interaction assay. 200 nM Alexa647-ATP (red pentagon) was mixed ATP, ADP, or ATP-γS (white pentagons) as a competitor at different concentrations, incubated with surface-immobilized 26S proteasome which was labeled with a SNAP-tag dye. The degree of Alexa647-ATP-proteasome colocalization in a steady state was measured using a TIRF microscope. The colocalization ratio (CR) after normalization by the competitor-free condition was inversely regressed on the competitor concentration to obtain the relative inhibitor constant $K_i$ (Materials and methods 'Single-molecule proteasome-nucleotide interaction assay'). ADP concentration is divided by 10 for presentation. The inset illustrates the experimental design. Error bars represent the standard deviation of three replicas. (**C**) Single-molecule translocation assay. N-terminal cyclinB was conjugated with Dylight550-labeled ubiquitins (yellow disks) on lysines (18, 36, and 64) and was incubated with surface-bound 26S proteasome in a buffer containing 0.5 mM ATP or with extra 0.8 mM ADP or with extra 40 μM ATP-γS. About 100 single-molecule traces exhibiting processive deubiquitylation in each condition were aligned by the time of substrate-proteasome encounter (t=0). The average fluorescent-ubiquitin intensity on a substrate molecule at each time point was calculated. The rate of translocation was calculated from the initial slope of the traces. The inset illustrates the experimental design. (**D**) Representative traces of the degradation of ubiquitylated cyclinB-iRFP by 1.5 nM purified 26S proteasome with ATP (rep1–3) or ADP in the buffer. (**E**) Lineweaver-Burk plot of the initial degradation rate ($v_0$) at varying concentrations of cyclinB-iRFP (Sub) either with 0.5 mM ATP (--) or with extra 0.8 mM ADP (+ADP). (**F**) The translocation rate of cyclinB-iRFP was measured using the fluorescent degradation assay with 0.5 mM ATP and various concentrations of ADP-Mg$^{2+}$. Error bars represent the standard deviation of 15 measurements. The red curve shows the prediction by the FEL model. FEL, free-energy landscape.

The online version of this article includes the following source data and figure supplement(s) for figure 3:

**Source data 1.** Degradation kinetics of ubiquitylated cyclinB-iRFP at various concentrations of substrate and proteasome.

**Source data 2.** Degradation kinetics of ubiquitylated cyclinB-iRFP in the presence of 500 μM ATP and various concentrations of ADP.

**Figure supplement 1.** A block diagram for simulating the dynamics of proteasome using the FEL model.

**Figure supplement 2.** Examples of single-molecule traces showing processive ubiquitin chain removal.

the basic chemical rate equations. The off-rate of a nucleotide is calculated from its $K_d$ as a function of the energy parameters, with the on-rate set at a constant (*Figure 3A*; Materials and methods 'Determining the FEL and kinetic parameters' and 'Determining the parameters related with substrate translocation'). The detailed process of a conformational change is not considered explicitly (i.e., as 'adiabatic') in that the change of nucleotide distribution alters the FEL and causes repopulation of the conformational space of proteasome. We do not include any assumption on the coordination of the nucleotide cycles, or the coupling of the nucleotide and conformational changes, or any predetermined sequence of transitions.

To generate testable predictions from simulated ATPase dynamics, we introduced a substrate peptide that is mechanically coupled with the continuous segment of the PL1 staircase. In these simulations, the disengaged PL1s occupy the top staircase position, as observed in cryo-EM structures (*Figure 1B*; *Dong et al., 2019*; *de la Peña et al., 2018*). The unit step of translocation is defined by the axial separation between PL1s, corresponding to approximately two amino acids (AAs) in the substrate polypeptide (*Figure 1B*). The force coupling with a translocating substrate may affect the rates of ATPase conformational changes. We represent this effect as a titling of the FEL and simplify by ignoring the stochastic and sequence-dependent variations of these forces and introduce two constant values to capture the average effect on the forward and backward processes (Materials and methods 'Determining the parameters related with substrate translocation').

To experimentally determine these two force parameters and another parameter for defining the activation energy barrier, we employed a quantitative degradation assay based on the decay of a fluorescent reporter of the ubiquitylated cyclinB N-terminus fused with an infrared fluorescent protein iRFP (*Figure 3D*; *Lu et al., 2017*). The N-terminal cyclinB is an unstructured protein that is efficiently degraded by the proteasome starting from its N-terminus (*King et al., 1996*; *Yamano et al., 2004*). We find that degradation of cyclinB-iRFP follows Michaelis-Menten-like kinetics with $K_M$=5.6 nM (*Figure 3E*). We performed each measurement at substrate concentrations much higher than the $K_M$ to maximize the signal-to-noise ratio, and tested five substrate concentrations, each with three replicas, to verify the saturation condition. It took an average of 55 s (i.e., the turnover time) for the proteasome to degrade a 46 kDa cyclinB-iRFP molecule in a process that could be rate-limited by several steps including substrate commitment, unfolding, and translocation (*Figure 3D*; *Collins and Goldberg, 2017*).

To determine the actual rate-limiting step, we engineered a mutant of cyclinB containing only three lysine residues, and conjugated fluorescent ubiquitin chains onto these lysines (*Lu et al., 2015*). We examined this substrate using a single-molecule translocation assay developed previously (*Figure 3C*; *Lu et al., 2015*; *Hon and Lu, 2019*). The processive translocation of a substrate coincides with the stepwise removal of entire ubiquitin chains by the deubiquitinase Rpn11 on the proteasome, which was detected by TIRF to measure the ubiquitin copies on a substrate molecule (*Figure 3—figure supplement 2*). The decay rate of fluorescent ubiquitins on three-lysine cyclinB suggests a translocation rate of 10.5 (±0.8) AA/s (Materials and methods 'Single-molecule proteasome assay'). A similar translocation rate was found in a measurement of $V_{max}$ of proteasomal degradation (*Peth et al., 2013b*).

With this translocation rate, a cyclinB-iRFP peptide containing 445 AAs would take at least 44 s to progress into the CP, that is, 80% of the 55 s total turnover time. This suggests that translocation is the limiting step of the entire degradation process for this substrate. Degradation of these substrates is unlikely to be limited by deubiquitylation because substrates with multiple Ub chains are degraded faster than the same substrate with fewer chains (*Lu et al., 2015*). Therefore, in the experiments described below, we used the measured degradation rate of cyclinB-iRFP as an approximation for the translocation rate. This reporter assay also gives consistent results with direct single-molecule measurements under perturbed conditions, such as in the presence of ADP or ATP-γS (*Figure 3C*). We next determined the translocation rate of cyclinB-iRFP in the presence of 500 µM ATP and different concentrations of ADP, and estimated the three translocation-related parameters using these data (*Figure 3F* and Materials and methods 'Determining the parameters related with substrate translocation').

## The FEL-predicted kinetics of substrate degradation

Directly probing the dynamical behavior of the proteasomal ATPases is challenging. If the FEL model can recapitulate the actual ATPase dynamics, it may contribute valuable insights into the functional mechanism of the proteasome. To evaluate the consistency of the model with reality, we seek to examine the predictions of the simulated dynamics in experiments that are independent of those for model construction, as discussed below. The workflow and the main results are summarized in *Figure 4—figure supplement 1*.

We found that the simulated ATPase dynamics at a steady state promotes a directional translocation of substrate into the CP, with variations due to the stochasticity of the dynamics (*Figure 4A*). We first compared the simulated translocation rates of cyclinB-iRFP at different ATP concentrations with

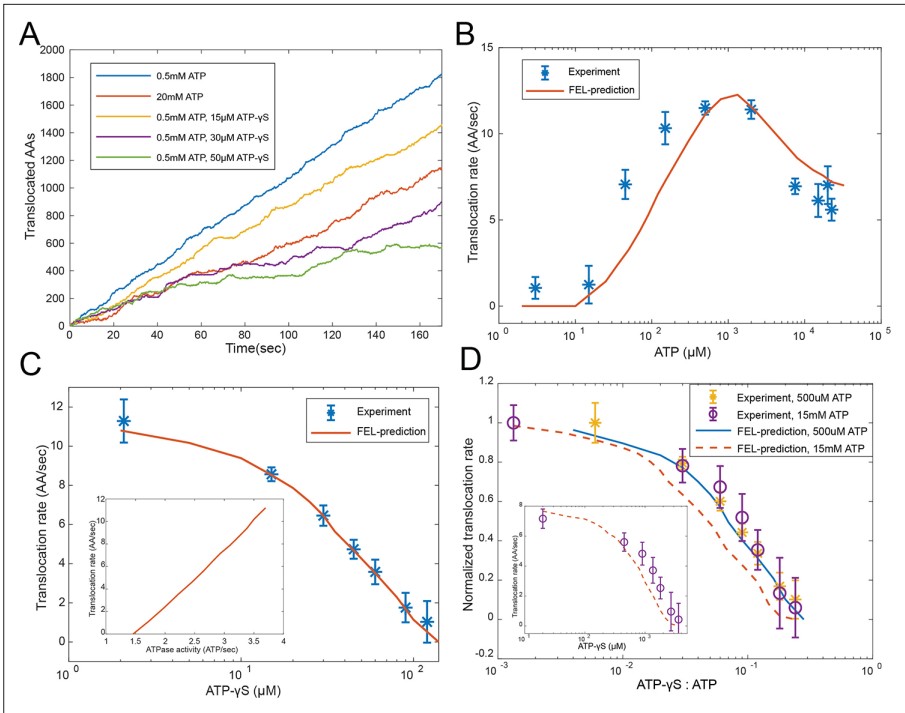

**Figure 4.** Evaluating the FEL-predicted degradation kinetics. (**A**) Examples of simulated kinetics of translocation on individual proteasome particles under indicated nucleotide conditions. (**B**) The translocation rate of cyclinB-iRFP measured at various concentrations of ATP-Mg$^{2+}$, in comparison with the FEL-model prediction (red curve) based on the determined parameters. (**C**) Same as in (**B**) but measured with 0.5 mM ATP and various concentrations of ATP-γS. Inset: translocation rate versus. ATP-hydrolysis activity as predicted by the FEL model at 0.5 mM ATP and different concentrations of ATP-γS. (**D**) Same as in (**B**) but measured with 15 mM ATP and various concentrations of ATP-γS. X-axis is the ratio between ATP-γS and ATP concentrations; Y-axis is the normalized translocation rate. The result in (**C**) is overlaid as a comparison. Inset: the absolute translocation rate versus ATP-γS concentrations. FEL, free-energy landscape.

The online version of this article includes the following source data and figure supplement(s) for figure 4:

**Source data 1.** Degradation kinetics of ubiquitylated cyclinB-iRFP in the presence of various concentrations of ATP.

**Source data 2.** Degradation kinetics of ubiquitylated cyclinB-iRFP in the presence of various concentrations of ATP.

**Source data 3.** Degradation kinetics of ubiquitylated cyclinB-iRFP in the presence of 500 μM ATP and various concentrations of ATP-γS.

**Source data 4.** Degradation kinetics of ubiquitylated cyclinB-iRFP in the presence of 15 mM ATP and various concentrations of ATP-γS.

**Figure supplement 1.** The workflow of this study, including the observations used for model construction, experimental validations of the simulated ATPase dynamics, and the major insights into the ATPase mechanism.

**Figure supplement 2.** High ATP concentration does not cause proteasome disassembly.

**Figure supplement 3.** High concentration of Mg$^{2+}$ alone does not affect the degradation kinetics.

**Figure supplement 4.** High ATP concentration does not lead to partial protein degradation or uncoupling between ubiquitylation and degradation.

**Figure supplement 5.** High ATP concentration does not inhibit the ATPase activity of proteasome.

experimental measurements (*Figure 4B*). Interestingly, the translocation-rate curve is non-monotonic and peaks around the physiological concentration of ATP. The FEL model accurately captures the quantitative features in both the up and down phases of the rate curve, each yielding a different insight. The EC$_{50}$ value for ATP in the up phase is 45 μM, far from the $K_d$ value of ATP or ADP at any binding pocket (*Figure 3A*: ~100 nM for group 1, ~3 μM for group 2, ~2 mM for group 3). Guided by the FEL model, we found that the expression of this EC$_{50}$ value is given by the ratio between the total

rate of ATP hydrolysis and the on-rate of ATP (Materials and methods 'Deriving a formula for the $EC_{50}$ value in the ATP titration experiment').

The observed inhibition of translocation at high ATP concentrations (*Figure 4B*) is likely due to the loss of ATPase cooperativity. This inhibition is not due to extra $Mg^{2+}$, proteasome disassembly, degradation-independent iRFP inactivation, or a general slowdown of ATP hydrolysis (*Figure 4— figure supplement 2* to *Figure 4—figure supplements 3–5*). As the FEL model suggests, coordinated movements of the ATPase subunits require the ATPase complex to have a unique energy-minimum conformation for each nucleotide status typically occurring during physiological operation, so that, when the nucleotide status changes, these ATPase's conformations undergo a well-defined collective change, leading to translocation. Very high ATP concentrations bias the proteasome toward all-ATP status and a flat FEL, resulting in a loss of ATPase cooperativity and abrogating the directionality of translocation (*Figures 2C and 4A*). This result is in contrast to the predictions of a sequential-transition model, which would predict a monotonic increase of translocation rate at increasing ATP concentrations, regardless of the parameters and reaction details, inconsistent with the observation (Materials and methods 'The translocation rate at different ATP concentrations as predicted by a strict sequential-transition model').

The FEL-predicted translocation kinetics are also consistent with the results of competition experiments involving ATP-γS. In the simulation (*Figure 4A*), ATP-γS introduces pauses between processive phases of substrate translocation, longer at higher concentrations, which closely resembles the ATP-γS-induced translocation pauses in single-molecule force measurements of ClpXP, a proteasome-like ATPase in prokaryotes (*Sen et al., 2013*). In the reporter experiment, 36 μM ATP-γS inhibited the translocation rate by 50% in the presence of 500 μM ATP (*Figure 4C*), despite the fact that the difference in the apparent $K_i$ values between ATP and ATP-γS is only twofold (*Figure 3B*). ATP-γS alters the ATPase dynamics which in turn affects the kinetics of nucleotide turnover. This process is required to interpret the low $IC_{50}$ for ATP-γS, as suggested by the FEL model.

We also performed an ATP-γS competition experiment at a level where the high-ATP-inhibition effect is apparent (15 mM ATP). Despite an overall reduction in translocation and a dramatic shift in $IC_{50}$, the FEL-predicted rates still closely match the experimental results (*Figure 4D*). The FEL model also predicts that the translocation rate should linearly depend on the rate of ATP hydrolysis at varying ATP-γS concentrations (*Figure 4C*), consistent with a previous observation (*Peth et al., 2013b*).

For structurally stable substrates, unfolding may be a limiting step in proteasomal degradation. To test the FEL model in the context of such substrates, we created a fluorescent reporter by inserting a dihydrofolate reductase (DHFR) domain from *Escherichia coli* between cyclinB and iRFP. We found that adding a DHFR ligand, folic acid, led to a dose-dependent stabilization of the ubiquitylated cyclinB-DHFR-iRFP in the presence of the proteasome, while the degradation of the original cyclinB-iRFP was unaffected (*Figure 5—figure supplement 1*). In the presence of folic acid, cyclinB-DHFR-iRFP degradation is still complete, or processive (*Figure 5—figure supplement 2*). The $IC_{50}$ value for folic acid in inhibiting the degradation is 800 μM, much higher than the 1 μM dissociation constant between DHFR and folic acid (*Posner et al., 1996*). This is likely because unfolding of the DHFR domain by the ATPase's actions primarily occurs when DHFR is transiently unliganded, consistent with the linear relationship between folic acid concentration and the inverse of the degradation rate (*Figure 5—figure supplement 3* and Materials and methods 'Interpreting the degradation kinetics of unfolding-limited substrates'). The FEL model suggests that folic acid lowers the overall efficiency of ATP utilization in degrading DHFR-containing substrates as reported previously (*Figure 5A*; *Peth et al., 2013b*). We tested the degradation rates of ubiquitylated cyclinB-DHFR-iRFP with 800 μM folic acid in an ATP-titration experiment, and found that folic acid did not affect the $EC_{50}$ value of ATP in the up-phase of the rate curve, although it lowered the peak degradation rate. However, folic acid significantly reduced the degree of degradation inhibition at high ATP concentrations (*Figure 5A and B*). This is likely because the ATPase activity unfolding the substrate is less affected by high ATP concentrations than is translocation (*Figure 4—figure supplement 5* and *Figure 5—figure supplement 4*). To predict the degradation rate of cyclinB-DHFR-iRFP, we introduced one additional parameter to describe the unfolding rate of unliganded DHFR by proteasome in the FEL model. These qualitative features of the predicted degradation-rate curve in the ATP-titration experiment are insensitive to the choice of this parameter (Materials and methods 'Interpreting the degradation kinetics of unfolding-limited substrates').

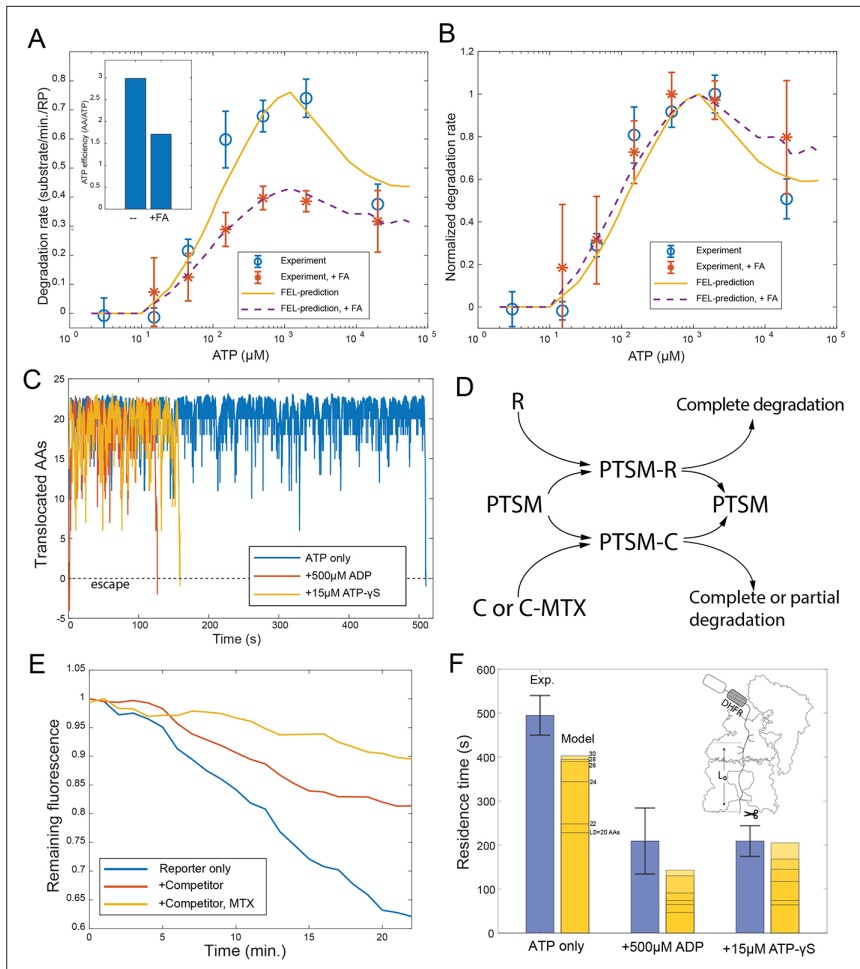

**Figure 5.** Evaluating the FEL predictions for structurally-stable substrates. (**A**) The initial degradation rate of ubiquitylated cyclinB-DHFR-iRFP by purified 26S proteasome with or without 0.8 mM folic acid (FA), overlaid with the FEL model prediction. The inset shows the effective length of substrate peptide degraded by consuming one ATP molecule, as predicted by the FEL model. The normalized degradation rate is shown in (**B**). Error bars represent the standard deviation of 15 measurements. (**C**) Examples of simulated translocation kinetics when the ATPases encounter an unfolding-resistant domain at $t=0$, assuming substrate escape occurs at $Y=0$. (**D**) Schematic showing the reactions in the competition assay for determining the residence time of an unfolding-resistant substrate on proteasome. R: cyclinB-iRFP reporter; C: cyclinB-DHFR-iRFP[dark] competitor. (**E**) Representative traces of the competition assay in (**D**). (**F**) Experimental values of the residence time of cyclinB-DHFR(MTX)-iRFP under indicated nucleotide conditions, in comparison with FEL model predictions with the peptide track length $L_0$ from 20 to 30 AAs. Error bars represent the standard deviation of 5 measurements. FEL, free-energy landscape.

The online version of this article includes the following source data and figure supplement(s) for figure 5:

**Source data 1.** Degradation kinetics of ubiquitylated cyclinB-DHFR-iRFP in the presence of various concentrations of ATP without folic acid.

**Source data 2.** Degradation kinetics of ubiquitylated cyclinB-DHFR-iRFP in the presence of various concentrations of ATP with folic acid.

**Source data 3.** Degradation kinetics of ubiquitylated cyclinB-iRFP in the presence of cyclinB-DHFR-iRFP[dark] as competitor, with either ADP or ATP-γS in the buffer.

**Figure supplement 1.** Folic acid slows down the degradation of cyclinB-DHFR-iRFP, but not cyclinB-iRFP.

**Figure supplement 2.** CyclinB-DHFR-iRFP was completely degraded by the proteasome in the presence of folic acid.

**Figure supplement 3.** The degradation rate of cyclinB-DHFR-iRFP is inversely proportional to the folic acid concentrations.

*Figure 5 continued on next page*

*Figure 5 continued*

**Figure supplement 4.** The rate of ATP hydrolysis versus ATP concentration as predicted by the FEL model.

**Figure supplement 5.** Sensitivity of the simulated translocation rates to parameter variations.

Molecular machines may occasionally run backward due to the stochasticity of single-molecular dynamics. Simulation by the FEL model suggests that substrates with an unfolding-resistant domain would not stably engage with the proteasome but would instead escape at a rate determined by the backward kinetics (*Figure 5C*). A stable ligand, such as methotrexate, can inhibit the degradation of cyclinB-DHFR-iRFP at a low concentration by preventing the unfolding of DHFR, resulting in partial cleavage of the substrate (*Figure 5—figure supplement 1*; *Johnston et al., 1995*). In a competition assay, degradation of the reporter cyclinB-iRFP was reduced by a nonfluorescent competitor cyclinB-DHFR-iRFP$^{dark}$ (*Figure 5D*). This inhibition was exacerbated in the presence of methotrexate (*Figure 5E*). We measured the turnover time of the stable DHFR substrate in this assay and found the value comparable with the model prediction which was calculated as a first-passage time on a 20–30 AA peptide track measured from the PL1s to the proteolysis sites in the CP (*Figure 5F*) (Materials and methods 'Monte Carlo simulation of the FEL model of proteasome' and 'Measuring the residence time of an unfolding-resistant substrate on the proteasome'). Adding ATP-γS or ADP in the simulation reduces the first-passage time and facilitates the escape of stable substrates (*Figure 5C*). These results are consistent with the experimental values for the turnover time in the presence of ATP-γS or ADP (*Figure 5F*).

In summary, we found that predictions from the FEL model closely match new experimental observations at nucleotide concentrations ranging across three orders of magnitude, in both the forward and the backward processes. The overall consistency with experiments is not sensitive to parameter uncertainty at a typical value of ~30% (*Figure 5—figure supplement 5*).

## Organization of proteasomal conformations in dynamical space

We further explore the features of the simulated ATPase dynamics and compare the predictions with additional results to examine this FEL approach. This analysis also provides insights into the mechanism of the proteasome and rationalizes the effects of ATPase mutations in previous studies.

Simulating the ATPase dynamics gives the steady-state distribution of the proteasome at each conformation and the frequency of every conformational transition. The FEL model is built on fundamental physical and chemical processes and does not a priori specify the occupancy of any conformation, the transition pathway between conformations, or how the ATPases cooperate. Interestingly, in the simulation, we found that the ATPase conformations self-organize into a transition network, with the $E_D$-like and $E_C$-like conformations predominating, as they do in the cryo-EM analysis (*Figure 6A*; *Dong et al., 2019*). Although the main transition pathway among $E_D$s resembles the previous sequential model, there are several side transitions, or branches, that are key to understanding some experimental observations.

To simulate the effects of an ATPase mutation, we abolished the ATP-hydrolyzing activity of one ATPase, for example Rpt3, in the simulation, and then analyzed the global dynamics of the mutated proteasome (*Miller and Enemark, 2016*). ATP hydrolysis at the Rpt3–Rpt4 pocket is important for the transition from the $E_{D1}$ to $E_{D2}$ cryo-EM state; its inactivation mimics a WB mutation and strongly reduces the transition frequency from the $E_{D1}$- to the $E_{D2}$-equivalent conformation in an FEL simulation. This loss of 'flow' was compensated by an increase in the transition to an $E_C$-like conformation, so that the overall translocation rate was only reduced by 11% (*Figure 6B* and *Figure 6—figure supplement 1*). Transition networks with an inactivated ATPase also show an expansion of the $E_D$-like conformations in which the ATPases at the –2 or –3 positions are flanked by open interfaces (0=mutant, negative=counterclockwise); this trend is consistent with and may explain a previous cryo-EM analysis on WB mutant proteasomes (*Eisele et al., 2018*).

The order of the elementary steps of reactions and transitions is an important part of the ATPase mechanism and is challenging to observe directly. Different sequences of these steps have been hypothesized to drive the orderly conformational transitions of the proteasome and other AAA+ ATPases (*Dong et al., 2019*; *Majumder et al., 2019*; *Monroe et al., 2017*; *Lyubimov et al., 2011*). Our simulation identifies the most-likely reaction sequence by enumerating all the possible sequences that can promote transitions between two states and calculating their probabilities. For the $E_{D1}$-to-$E_{D2}$

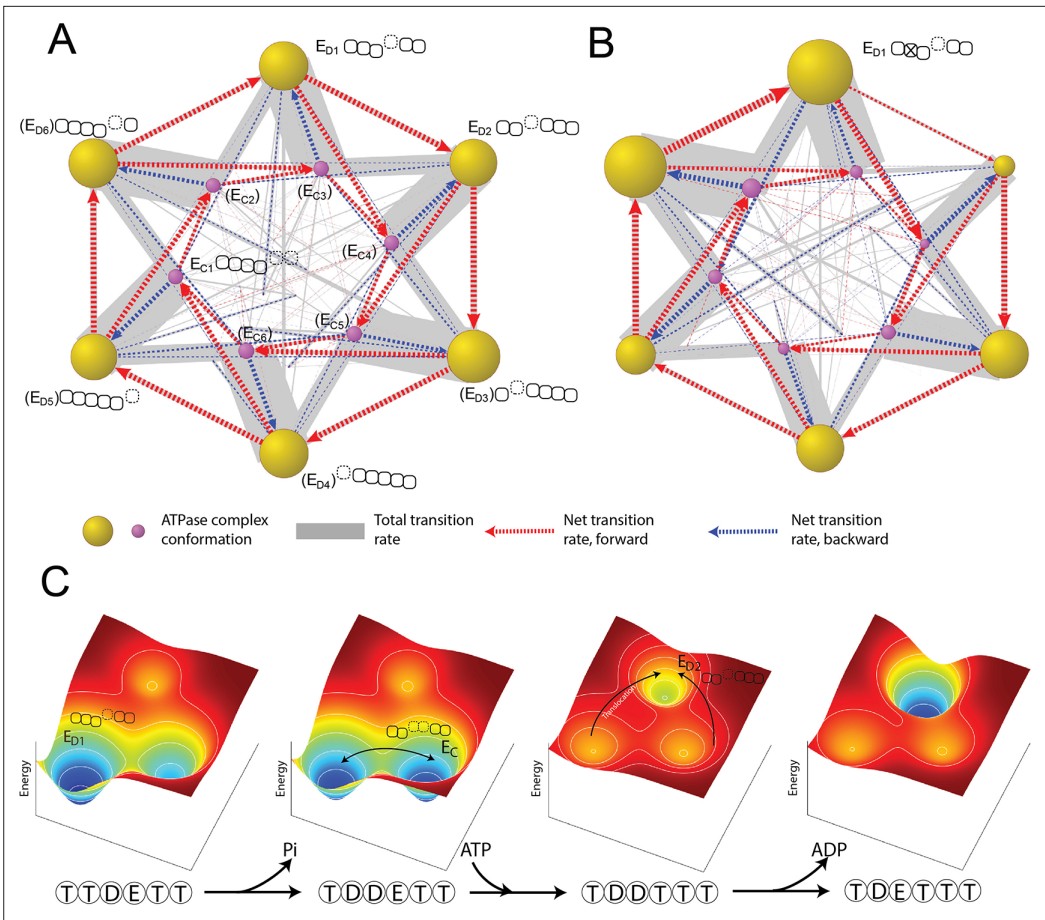

**Figure 6.** Organization of proteasomal conformations in dynamical space. (**A**) A diagram showing the steady-state transitions between the ATPase complex conformations in the FEL model. Each node represents a unique hexamer conformation, whose size is in proportion to its probability of occupancy in a steady state. Yellow and purple nodes are the $E_D$-like and $E_C$-like conformations, respectively, with the corresponding ATPase architecture as presented in *Figure 2C*. The labels of the conformations that have not yet been experimentally identified are bracketed. The width of a gray line is proportional to the total rate of forward and backward transitions between two conformations; and a dashed arrow represents the net transition rate and is colored according to the direction of substrate translocation associated with this conformational change. Minor conformations are randomly distributed in the graph. ***Supplementary file 1*** contains a full list of the conformations and their transition rates. The transition diagram without the hydrolysis activity of Rpt3 is shown in (**B**). (**C**) The mechanism of the cooperative movements of proteasomal ATPases during substrate translocation. The FELs on three conformations (the $E_{D1}$- and $E_{D2}$-equivalent and an $E_C$) are illustrated for four nucleotide statuses in a typical $E_{D1}$-to-$E_{D2}$ transition (***Figure 6—figure supplement 2***). The ATPase architecture of each conformation is presented as in ***Figure 2C***. Arrows indicate the major conformation transitions upon the change of the nucleotide status and the associated FEL. FEL, free-energy landscape.

The online version of this article includes the following figure supplement(s) for figure 6:

**Figure supplement 1.** The rates of substrate translocation and ATP hydrolysis for a Walker-B mutant proteasome.

**Figure supplement 2.** The typical sequence of elementary steps underlying the $E_{D1}$-to-$E_{D2}$ state transition.

**Figure supplement 3.** The typical sequence of elementary steps underlying the $E_{D1}$-to-$E_C$ state transitions.

**Figure supplement 4.** The Lid-ATPase interaction may account for the observed dissymmetry in conformational occupancies in previous cryo-EM studies.

**Figure supplement 5.** The Lid-ATPase interaction may account for the different growth phenotypes of yeast Walker-B mutants.

**Figure supplement 6.** Global dynamical space with heterogeneous ATPase parameters.

transition, the most-likely reaction sequence begins with ATP hydrolysis at the Rpt3–Rpt4 pocket, followed by ATP binding to Rpt5 and a conformational rearrangement of the ATPase complex. This conformational change exchanges the ADP-bound Rpt3–Rpt4 interface for an ATP-bound Rpt5–Rpt1 interface and drives a unit-step translocation of two AAs by rearranging the PL1 staircase (*Figure 6— figure supplement 2*). In this process, 1.6 kcal/mol energy is dissipated. This energy-dissipating step is important for establishing the directionality of translocation in the simulation. This conformational change also drives Rpt4 into a disengaged APO state with a weak nucleotide affinity, allowing rapid release of bound ADP. Otherwise, the rate of ADP release would be incompatible with the fast kinetics of substrate translocation. Phosphate release is unlikely to be limiting since phosphate has a $K_i$ value of 70 mM measured in a competition assay, a much weaker affinity than ADP for the ATPases (*Figure 2— figure supplement 1*).

Identifying this 'most-likely' reaction sequence is critical for obtaining the formula for the $EC_{50}$ value of ATP (Materials and methods 'Deriving a formula for the $EC_{50}$ value in the ATP titration experiment'). In a proteasome with non-hydrolyzing Rpt3, the $E_{D1}$-to-$E_C$ transition that partially rescues the 'flow' is initiated by ATP hydrolysis at the Rpt6–Rpt3 pocket. The associated conformational change drives a translocation of two unit-steps, or four AAs (*Figure 6—figure supplement 3*). These larger steps occur stochastically even in the wild-type (WT) proteasome, though they are less energetically favorable. Nonetheless, these occasional larger steps likely explain the experimentally observed translocation efficiency of 2.6–3.0 AAs per ATP consumption (*Figure 5A*; *Peth et al., 2013b*).

We next address the question of whether our model can help to interpret the observed functional disparity among the proteasomal ATPases. The model described above is symmetric, with all ATPases having identical parameter. One potential source of symmetry-breaking is the proteasome Lid-ATPase interaction. The Lid subcomplex interacts extensively with the ATPase domains, primarily on Rpt3/6, in the $E_A$-like or $E_{A'}$-like states in which the PL1s on Rpt3 and Rpt2 respectively occupy the top and bottom niches in the staircase and Rpt6 is in a disengaged position (*Dong et al., 2019*; *Chen et al., 2016*; *Zhu et al., 2018*; *Beck et al., 2012*). This arrangement closely resembles one $E_D$-like conformation (*Figure 6—figure supplement 4*, gray node). Weakening the Lid-ATPase interaction by mutations reduces the fraction of proteasome in the $E_A$-like states (*Greene et al., 2019*). We therefore hypothesize that the Lid-ATPase interaction may effectively stabilize this specific ATPase conformation by lowering its standard free energy. We implemented such a stabilizing effect in the FEL simulation by lowering the free energy of this $E_D$-like conformation by an arbitrary value, and found that this resulted in significant expansion of the occupancy of the $E_{D1}$- and $E_{D2}$-conformations and a simultaneous shrinkage in other $E_D$-like conformations (*Figure 6—figure supplement 4*), consistent with the cryo-EM observations (*Dong et al., 2019*; *de la Peña et al., 2018*).

To study whether the Lid-ATPase interaction may contribute to the different growth phenotypes in yeast WB mutants, we examined how the translocation rate changed after simulated inactivation of the ATP-hydrolysis activity of individual ATPases in the presence of the Lid. We found that the predicted translocation rate is generally correlated with the corresponding growth phenotype in a range of the Lid-ATPase interaction strength, which suggests that this interaction may contribute to the growth phenotypes of yeast WB mutants, potentially in conjunction with other compensatory mechanisms (*Figure 6—figure supplement 5*).

## Discussion

Protein machines accomplish complex tasks, driven by their exquisite structural dynamics at the molecular level. Studying these dynamics is challenging for both experimental investigation and molecular simulation.

In this work, we exercised the principle of parsimony to reconstruct the FEL of the proteasomal ATPase complex and experimentally determined its parameters in an attempt to uncover the mechanism in polypeptide translocation. The FEL is an intrinsic property of a protein or a complex and is a key component of physical models that have provided important insights into the mechanisms of molecular motors (*Wang and Oster, 1998*; *Tu and Cao, 2018*; *Bustamante et al., 2001*). FELs can be derived from conformational occupancies in cryo-EM studies through the Boltzmann equation *Dashti et al., 2014*; *Fischer et al., 2010*. Identification of the very large number of proteasomal states is challenging and the nonequilibrium transitions further complicate the analysis. The FEL-based simulation introduced in our study generates various predictions which we used to examine the validity of

the simulated ATPase dynamics and this overall approach. We found that the simulation recapitulated a number of important experimental observations including degradation kinetics, state distributions in cryo-EM datasets and the growth phenotypes of WB mutants (*Figure 4—figure supplement 1*). Moreover, this study reveals that these varieties of phenomena are in fact driven by a coherent and simple principle embodied in our model, thus offering mechanistic insights into the ATP-driven cooperative actions of the ATPases in promoting the translocation of substrate polypeptides.

The observation that the nucleotide status of three consecutive ATPase subunits on the proteasome represents a continuous sequence in an ATP cycle has led to a sequential rotary model and 'hand-over-hand' translocation for proteasomal ATPase activity (*Puchades et al., 2020*; *Dong et al., 2019*; *de la Peña et al., 2018*). A strict sequential mechanism requires that the ATP-bound subunit adjacent to the ADP subunit must hydrolyze ATP first. We compared the nucleotide-interacting motifs in all high-resolution structures of the proteasome but failed to identify a consistent trend that could suggest an allosteric effect between the ATP-pocket and other parts of the complex. As an alternative, we considered the possibility that the apparent cooperativity between different subunits might emerge from the modulation of their collective FEL by ligand binding, as proposed in the celebrated Monod-Wyman-Changeux concerted model for allosteric regulation (*Changeux, 2012*). Loss of ATPase cooperativity due to flattening of the FEL at above-physiological ATP concentrations reduces the rate of substrate degradation (*Figure 4B*). A similar inhibitory effect has been reported for a bacterial pilus assembly motor, though the exact mechanism is still unclear (*Sukmana and Yang, 2018*). Although sacrificing some translocation efficiency, the flat FEL of the all-ATP state facilitates interchange among proteasomal conformations, even under physiological ATP concentrations. This property may be important for bypassing an occasional stuck or defective ATPase subunit that would completely inhibit proteasomal activity in a strict sequential-transition model. The resulting network of conformational transitions is analogous to the 'ring-resetting' model for the ClpXP ATPase complex (*Stinson et al., 2013*). It is interesting to note that the peptidase activity of the proteasome is also inhibited by ATP concentrations above a threshold; however, this is likely to be due to a different mechanism since the critical concentration for peptidase inhibition is ~50× lower than that for translocation (*Smith et al., 2011*).

One or two ATPase units are disengaged from substrate peptide or from the translocation channel in all identified states of the proteasome. These disengaged ATPases are flanked by two open interfaces except in $E_A$ states, where the 'disengaged' subunit is associated with one open interface. We speculate that this is due to the extensive Lid-ATPase interaction in these states (*Chen et al., 2016*). The presence of disengaged ATPases ensures that there is a unique energy-minimum conformation, and thus ensures a high level of cooperativity among the ATPase subunits. The smaller pocket energy $E_p$ of these disengaged units also allows a fast nucleotide exchange, which is required for efficient translocation. Disengaged ATPase units are frequently observed in the structures of ring-shaped ATPases and may have a similar role in their activities (*Majumder et al., 2019*; *Fei et al., 2020*; *Puchades, 2017*).

The nonequilibrium transitions of the ATPase complex identified by the FEL approach reveal important features of proteasome functions. Our model shows that substrate translocation steps are directly coupled to an energy-dissipating conformational transition which swaps an ADP-bound closed interface with another ATP closed interface (*Figure 6—figure supplement 2*). We propose that this coupling may dictate the directionality of translocation. The simulated translocation process deviates from a strict 'hand-over-hand' mechanism in that the step size in our model can adopt multiples of 2× AA, depending on which ATP molecule is first hydrolyzed (*Figure 6—figure supplement 2* and *Figure 6—figure supplement 3*). Variations in step size have been observed in substrate translocation by ClpXP, though its connection with our simulation is unclear (*Sen et al., 2013*; *Shi et al., 2016*).

We used the cryo-EM structures of proteasome as the basis for model construction. These structures are mainly for identifying the DOFs of the ATPase's conformational change (*Figure 4—figure supplement 1*). Other aspects of the structural information, such as the conformational occupancy and nucleotide distribution, are not required for model construction but are still consistent with its predictions. The inter-subunit signaling motif has been suggested to mediate ATPase communication and coupling with nucleotides and may contribute to the bridge energy in the FEL model (*Chang et al., 2017*). We did not include explicit allosteric coupling between ATPases in our model, as we find that it is not essential for explaining the current experimental observations. Adding allosteric effects

to the model would be straightforward and may accommodate the 'coordinated bursts' mechanism to extend this model to other ATPase systems (**Fei et al., 2020**).

Partial degradation, or processing, by the proteasome is a natural process in the maturation of certain protein factors (**Nassif et al., 2014**). The backward process on proteasome is much less understood and may contribute to efficient release of partially degraded substrates. In our study, modeling the substrate-escape kinetics as a first-passage time problem yields predictions consistent with the measurements. Other mechanisms, such as loss of grip on certain substrates, may also contribute to the escape kinetics (**Nassif et al., 2014**). In the current model, we simplified the mechanical force on a substrate during translocation as a constant. Incorporating the precise unfolding landscape of a substrate and the interaction energy with the ATPase into the FEL model would be straightforward (**Saha and Warshel, 2021**), and may shed light on the mechanism of partial or nonprocessive degradation of certain substrates.

The same mutation in the six highly-similar ATPases often have different effects on proteasome activities (**Rubin et al., 1998**; **Eisele et al., 2018**; **Beckwith et al., 2013**). Previous studies lead to no consensus on whether certain subunits play a more important role in substrate degradation and the extent to which the sequence divergence among the six ATPases contributes to these functional heterogeneities is unclear. Interestingly, we found that it is possible to rationalize at least some of these functional heterogeneities without invoking disparity at the level of individual ATPases. The introduction of Lid-ATPase interactions as a simple conformation-stabilizing parameter in the FEL simulation, recapitulates the asymmetric cryo-EM state distributions and may explain the different phenotypes of WB mutants. Direct interpretation of WA mutants remains challenging, as these mutations often lead to proteasome assembly defects (**Beckwith et al., 2013**). In the simulation, large heterogeneities of energy parameters among the six ATPases can alter the ATPase dynamics (**Figure 6—figure supplement 6**). Interestingly, this effect is significantly blunted by incorporating the Lid-ATPase interaction (**Figure 6—figure supplement 6**). This may contribute to the inessentiality of parameter variations in recapitulating the experimental results. Including the Lid-ATPase interaction in the model yields predictions that are compatible with all the kinetic measurements in our study and indicates that the proteasome may occasionally visit an $E_A$-like state during translocation. However, the peak translocation rate is underestimated by 20%–25% in this case, suggesting that our understanding of the symmetry-breaking mechanism is still incomplete.

The FEL presented here is undoubtedly an approximation of the actual FEL of the proteasome. Results in future studies may identify additional mechanisms, such as allosteric effects, to increase the accuracy of the simulated ATPase dynamics. As one example of the results that are not well recapitulated by the current model, protein substrates and ubiquitin chains activate the ATPase activity of the proteasome by ~2-fold, larger than the predicted 9% increase by the FEL model (**Peth et al., 2013a**). This activation step may be independent of the processive translocation phase described by the FEL model, and may involve a transition from the resting to translocating states of the proteasome (**Bard et al., 2019**). In the $E_A$ states of the proteasome, all ATPases bind nucleotides, suggesting that the disengaged subunit in the resting states may have higher nucleotide affinity than the disengaged ones in the translocating states (**Dong et al., 2019**). This high affinity may limit nucleotide turnover in the resting states. Similarly, our model may not accurately describe the behaviors of substrates whose degradation is limited by steps other than translocation, such as the DHFR-containing substrates in this study. Extending the FEL model to cover such substrates may reveal other important aspects of the proteasome's activities.

We expect this FEL approach to have many applications in guiding the experimental design and data analysis, and to provide valuable insights into the mechanism of proteasomal ATPases and other AAA machines.

## Materials and methods

### Key resources table

| Reagent type (species) or resource | Designation | Source or reference | Identifiers | Additional information |
|---|---|---|---|---|
| Recombinant DNA reagent | pT7-CyclnB-iRFP (plasmid) | This study | pLM254 | For expressing cyclinB-iRFP (available upon request) |

*Continued on next page*

| Reagent type (species) or resource | Designation | Source or reference | Identifiers | Additional information |
|---|---|---|---|---|
| Recombinant DNA reagent | pT7-cyclinB-DHFR-iRFP (plasmid) | This study | pLM428 | For expressing cyclinB-DHFR-iRFP (available upon request) |
| Recombinant DNA reagent | pT7-cyclinB-iRFP-DHFR (plasmid) | This study | pLM429 | For expressing cyclinB-iRFP-DHFR (available upon request) |
| Recombinant DNA reagent | pT7-cyclinB (K18, 36, 64) (plasmid) | This study | pLM120 | For expressing cyclinB (K18, 36, 64) (available upon request) |
| Peptide, recombinant protein | Dylight550-Ubiquitin | *Lu et al., 2015*; *Puchades et al., 2020* | Dy550-Ub | Available upon request |
| Peptide, recombinant protein | Human 26S proteasome | HEK293 cell (Rpn11-HTBH) | hPTSM | |
| Peptide, recombinant protein | Human 26S proteasome SNAP-Rpn3 | HE293 cell (SNAP-Rpn3) This study | hPTSM-SNAP | |
| Cell line (Human) | HEK293 | Lab stock (commonly available) | HEK293 | |
| Cell line (Human) | HEK293-SNAP-Rpn3 | This study | HEK293-Rpn3-SNAP | For expressing SNAP-Rpn3 proteasome |
| Chemical compound, drug | ATP-gS | Sigma-Aldrich | A1388 | |
| Chemical compound, drug | A647-ATP | Thermo Fisher Scientific | A22362 | |
| Chemical compound, drug | Folic acid | Sigma-Aldrich | F8758 | |
| Chemical compound, drug | Methotrexate | Sigma-Aldrich | A6770 | |
| Chemical compound, drug | Biliverdin | Sigma-Aldrich | 30891 | |
| Chemical compound, drug | SNAP-surface-549 | NEB | S9112S | |
| Strain, strain background (*Escherichia coli*) | NiCo21 DE3 | NEB | C2529H | |
| Software, algorithm | MATLAB 2018 | MathWorks | | |
| Software, algorithm | Pajek | Pajek | http://vlado.fmf.uni-lj.si/pub/networks/pajek/ | |
| Software, algorithm | Proteasome FEL model | This study | https://github.com/luyinghms/Proteasome-FEL-model.git; *Ying, 2022* | Source code for the FEL model |
| Chemical compound, drug | ATP-gS | Sigma-Aldrich | A1388 | |

## Analysis of cryo-EM structures

The nucleotide-binding pockets were defined based on the structure of $E_{A1}$ state. Briefly, for each Rpt subunit of $E_{A1}$ state, the amino acids within 6–9 Å of the bound nucleotide were selected and defined as the nucleotide-binding pocket. Structures of each nucleotide-binding pocket in different states of

the proteasome were aligned in Pymol and the RMSD of the aligned atoms in the nucleotide pocket was reported.

To gauge the interface between two ATPase subunits, the SESA was used to report the buried area formed by the interacting residues. Using Pymol, the solvent-accessible surface area (SASA) for individual ATPase domains in isolation and the SASA value for the complex were calculated. Then the SESA between two ATPase domains was calculated using this formula (using Rpt1 and Rpt2 as an example): SESA of Rpt1-Rpt2=[(SASA of Rpt1)+(SASA of Rpt2)–(SASA of Rpt1-Rpt2 complex)]/2.

## Protein purification and labeling

Recombinant human N-terminal (1–88) cyclinB1 (cycB_NT), 3-lysine cycB_NT (K18, K36, and K64), cycB_NT-iRFP, cycB_NT-DHFR-iRFP were purified from *E. coli* cells using a polyHis tag and Ni$^{2+}$-affinity column. PKA (protein kinase A) sites (RRASV) was placed at both the N- and C-terminus of cycB_NT-iRFP and cycB_NT-DHFR-iRFP for detection by autoradiography in degradation assays.

Human ubiquitin with a cysteine residue and a polyHis-tag at the N-terminus was purified from *E. coli* cells using cation exchange chromatography (GE, 17-1152-01) and was labeled with Dylight-550-maleimide (Pierce, 62290). After removing unreacted dyes, labeled ubiquitin was subjected to anion exchange chromatography (GE, 17-1153-01) to separate labeled and unlabeled ubiquitin. Finally, the N-terminal polyHis tag was cleaved off using thrombin.

Anti-20S antibody (MCP21) was biotinylated using biotin-NHS (Pierce, 20217), and was purified using a desalting column.

Radioactive (*Lyubimov et al., 2011*) p-ATP was used to label substrates with a PKA site at the N-terminus for in vitro ubiquitylation and degradation assays.

Human E1, E2 UbcH10, WT-ubiquitin were purchased from BostonBiochem. Purified streptavidin was from Invitrogen.

Protein concentrations were determined using Bio-Rad protein assay; ubiquitin concentration was determined by UV A280 absorption.

## Recombinant anaphase-promoting complex/cyclosome (APC)-Cdh1 purification

Purification of recombinant APC-Cdh1 from insect cells has been described elsewhere (*Brown et al., 2016*; *Weissmann et al., 2016*). Briefly, viruses expressing 14 APC components were generated by transfecting Sf9 insect cells with the recombinant baculoviral genome based on a biGBac system using Fugene 6 reagent. Amplified viruses were added to HighFive insect cell culture for protein expression.

APC was expressed with a Twin-Step(II)-tag on the C-terminus of APC4, and was isolated from cell lysate using Strep-Tactin sepharose, and then was polished by ion-exchange chromatography and gel filtration. Myc-6xHis-Cdh1 was purified from HighFive cells using Ni$^{2+}$ agarose.

## Affinity purification of the human 26S proteasome

Human proteasomes were affinity-purified on a large scale from a stable HEK293T cell line harboring HTBH tagged hRPN11 (a gift from L. Huang). The cells were Dounce-homogenized in a lysis buffer (50 mM NaH2PO4 [pH7.5], 100 mM NaCl, 10% glycerol, 5 mM MgCl2, 0.5% NP-40, 5 mM ATP, and 1 mM DTT) containing protease inhibitors. The lysate was cleared and incubated with NeutrAvidin agarose beads (Thermo Fisher Scientific) overnight at 4°C. The beads were then washed with excess lysis buffer followed by a wash buffer (50 mM Tris-HCl[pH7.5], 1 mM MgCl2, and 1 mM ATP-Mg$^{2+}$). Usp14 on proteasome was washed off using the wash buffer plus 100 mM NaCl for 30 min. 26S proteasomes were eluted from the beads by cleavage using TEV protease (Invitrogen).

For purifying SNAP-tagged proteasome, the HTBH-Rpn11 cell line was stably transfected with a lentivirus carrying FLAG-SNAP-Rpn3 under a CMV promoter, and the SNAP-proteasome was purified from the transfected cell line using the above procedure.

Identity of the cell lines is authenticated using STR profiling and mycoplasma contamination was regularly tested by PCR.

### In vitro ubiquitylation reaction

The APC ubiquitylation reactions were carried out in the UBAB buffer (25 mM Tris-HCL[pH 7.5], 50 mM NaCl, and 10 mM MgCl$_2$) containing 30 nM APC-Cdh1, 100 nM E1, 2 µM UbcH10, 2 mg/ml BSA, the energy regenerating systems, 1 µM substrate (cycB_NT, cycB_NT-iRFP, or cycB_NT-DHFR-iRFP), and 100 uM WT-ubiquitin or 15 µM Dylight550-ubiquitin, incubated for 4 hr at 25°. In case, the substrates were (*Lyubimov et al., 2011*) p-labeled, calyculin A (EMD, 19–139) was added at 10 µg/ml to prevent dephosphorylation.

### iRFP substrate degradation assay

Ubiquitylated iRFP substrates were diluted in a buffer containing 1× UBAB, 10% glycerol, 1 mM DTT, 0.5 ml/ml γ-globulin (Sigma-Aldrich, G5009), 0.05% NP-40, and nucleotide-Mg$^{2+}$ at the experimental concentrations, and were aliquoted to a 384-well plate (Corning 3544). Purified human 26S proteasome was added at 1.0–3.0 nM final concentration to start the reaction. Degradation kinetics was monitored using a microplate reader (BioTek Synergy H1) once every 90 s at 35°.

For each buffer condition, the degradation measurement was performed at five substrate concentrations (40 nM, 60 nM, 80 nM, 120 nM, and 200 nM), with three replicas for each concentration. A standard reaction with 500 µM ATP-Mg$^{2+}$ was included in every batch of reactions to control for sample batch-to-batch variations.

### Data analysis

The initial degradation rate ($v_0$) was obtained using a linear fitting of the degradation curve in the first 15 min after temperature stabilization. The turnover time is the inverse of the degradation rate divided by the proteasome concentration.

### Single-molecule proteasome translocation assay

The detailed procedure of single-molecule proteasome assay has been described before (*Lu et al., 2015*; *Hon and Lu, 2019*). Briefly, 15 nM 26S proteasome and 15 nM biotinylated MCP21 antibody were mixed and incubated at room temperature for 15 min. The proteasome-antibody mix was loaded onto PEG-passivated slides coated with streptavidin. After a brief incubation, unbound protein was washed off, and was exchanged into an imaging buffer (1× UBAB, 20 mM Imidazole, 2 mg/ml BSA, and nucleotides) containing diluted ubiquitylated substrate at ~1 nM.

Images were acquired at 100 ms per frame on a custom TIRF microscope equipped with three laser lines of 488 nM (150 mW), 561 nM (150 mM), 638 mM (100 mW), and a Pco SCMOS camera. The single-molecule experiment was performed at room temperature of ~27°.

### Data analysis

The basic image processing and single-particle identification were performed as described previously (*Lu et al., 2015*; *Hon and Lu, 2019*). The 'stepped' traces as in *Figure 3—figure supplement 2* which signaled the co-translocation deubiquitylation mediated by Rpn11 were identified and aligned by the time of substrate-proteasome interaction. The average intensity of fluorescent ubiquitins among these 'stepped' traces was calculated for each time point, and the rate of translocation was calculated using a linear fitting of the averaged trace between 1 and 3 s after proteasome-substrate interaction. Data analysis was carried out in Matlab 2018.

### Single-molecule proteasome-nucleotide interaction assay

The coverslip surface was passivated with Tween20 and functionalized with biotinylated BSA as described previously (*Hua et al., 2014*). 50 nM SNAP-Rpn3 proteasome was incubated with 15 nM biotinylated MCP21 antibody and 1 uM SNAP-surface 549 dye for 30 min at room temperature. The proteasome-antibody mix was buffer-exchanged into buffer W (1× UBAB, 0.05% Tween20, and 0.5 mM DTT) + 0.4 mg/ml BSA using a 30-kD concentrator, and then was buffer-exchanged into buffer W + 1 µM Alexa647-ATP at 4°. The sample was diluted by five times in an imaging buffer (buffer W + competitive nucleotides + 200 nM Alexa647-ATP (Thermo Fisher Scientific) + PCA/PCD as the oxygen scavenging system) and was incubated for 20 min on ice before loading onto the passivated surface via streptavidin. Images acquisition started after a 3-min incubation. We did not observe obvious 19S–20S dissociation as suggested by the fluorescent SNAP-Rpn3 signal in a 30-min incubation.

## Data analysis

The basic image processing and single-particle identification were performed as described previously (*Lu et al., 2015*; *Hon and Lu, 2019*). We performed colocalization (<1 pixel) analysis of the SNAP-tag signal with Alexa647-ATP. The fraction of SNAP-tagged proteasome particles that interacted with Alexa647-ATP under a steady-state condition was calculated as the colocalization ratio (CR). The inhibitor constant $K_i$ value for each type of nucleotide was obtained by a linear regression using the following formula:

$$C.R = \frac{[ATP^*]}{K_{ATP^*}\left(1+\frac{[NT]}{K}\right)+[ATP^*]} \qquad (1)$$

CR: colocalization ratio; [ATP*]: concentration of Alexa647-ATP; $K_{ATP^*}$: dissociation constant of Alexa647-ATP; [NT]: concentration of competing nucleotide. Data analysis was performed in Matlab 2018.

## Constructing the nucleotide-dependent free-energy landscape (FEL) of the ATPase complex on proteasome to simulate its conformational dynamics

### Basic definitions and assumptions

- Conformation: A conformation here is narrowly defined as a geometric arrangement of the six ATPases, regardless of their bound nucleotides. Totally 30 conformations are included in the FEL (Materials and methods 'Defining a discrete conformational space of the proteasome ATPase complex by extrapolating the cryo-EM observations').
- State: Unless specified in the context, a state here refers to a unique combination of an ATPase conformation and a nucleotide distribution in the six binding pockets. Since each nucleotide pocket can be empty, ATP-bound, ADP-bound, or ATP-γS-bound, the total number of states in the nucleotide-dependent FEL is $30*4^6$=122,880.
- Dynamical space: a complete description of the conformational changes of the proteasomal ATPase complex, including the conformations, the conformational occupancies, the conformational transition rates, and their dependence on the nucleotide distribution.

We assumed that the six ATPases on proteasome were associated with the same set of energy and kinetic parameters. The substrate peptide was assumed to be tightly gripped by the pore-1 loops (PL1) on the ATPases without slippage during translocation.

### Defining a discrete conformational space of the proteasome ATPase complex by extrapolating the cryo-EM observations

We designated each conformation in the FEL by the states of its six ATPase domain interfaces, either open or closed. We constrained the total number of open interfaces in each conformation to be either 2 or 3, since only two or three open interfaces have been observed in the cryo-EM states of the proteasome. Five conformations having 3+3 or 2+2+2 symmetry in the ATPase architecture were excluded to avoid ambiguity in assigning substrate-PL1 interaction. After all, the FEL contains a total of 30 discrete conformations (*Supplementary file 1*). Four conformations have so far been identified by the cryo-EM structural analysis.

Based on the cryo-EM observation, we incorporated the following rules to simulate substrate translocation: a series of closed interfaces positioned the PL1s on consecutive ATPases into a staircase arrangement; the axial separation between adjacent PL1s was 2× AAs. Open interfaces did not constrain the relative geometry between adjacent ATPases. If one or two consecutive ATPase subunits were flanked by open interfaces at both ends, they were disengaged from substrate interaction, and their PL1s moved to the top registry of the staircase (Note 1). Disengaged ATPases were assigned with a different set of energy parameters to reflect the *APO* configuration of their nucleotide *cis* pockets in cryo-EM structures. In the scenarios involving three ATPase segments, the largest segment was assumed to interact with the substrate. The distance between the 20S and the PL1 at the lowest staircase position was considered invariant. The distance of substrate translocation of each conformational change is taken as the difference between the average staircase positions of those PL1s that interact with the substrate both before and after the transition.

*Note 1: As long as these ATPases are disengaged from substrate interaction and do not block the movement of other ATPases, whether they stay at the top registry or not does not affect the simulation result.

## Determining the FEL and kinetic parameters

1. $k_{on}$=1e5/M/s for all the ATPase pockets and nucleotides. No measurement of the nucleotide $k_{on}$ for proteasome has been reported in our record. To estimate the upper bound of $k_{on}$, we use the observation that ADP-filled interfaces in cryo-EM structures had approximately equal tendency to be open or closed, which requires $e_{Br,ADP}$ ~0 (±kT) (*Figure 1C*); this condition can only be achieved if $k_{on}$<3e5/M/s. To estimate the lower bound, we consider that the proteasome can still translocate and degrade substrates efficiently at ≤100 µM ATP (*Figure 4B*). This requires $k_{on}$>2e4/M/s, otherwise nucleotide rebinding would consume too much time, inconsistent with the fast translocation kinetics. Therefore, we estimated $k_{on}$ as the geometric mean of its upper and lower bounds. This on-rate estimation is consistent with that calculated directly from the ATP EC50 expression (*Figure 4B* and Materials and methods 'Deriving a formula for the EC$_{50}$ value in the ATP titration experiment'). This value is also consistent with the nucleotide $k_{on}$ for F1-ATPase, in spite of its divergence from proteasomal ATPases (*Al-Shawi and Nakamoto, 1997*).

2. $e_p^{APO}$=−3.7 (kcal/mol) for ATP, ADP, and ATP-γS at an *APO* pocket. We noticed that the disengaged ATPases were nucleotide-bound in the yeast proteasome cryo-EM structures despite high similarity with the human proteasome conformations (*Dong et al., 2019*; *de la Peña et al., 2018*). This, we surmise, is because 5 mM ATP was used in the yeast proteasome study while 1 mM ATP was used in our human proteasome project. In addition, higher-than-2mM concentrations of ATP-γS have been found to alter the proteasome conformational distribution in a bulk measurement, which should be due to the nucleotide interacting with a weak pocket that has a comparable $K_d1$ (*Erzberger and Berger, 2006*). Therefore, we set the $K_d$=2 mM for an *APO* pocket, which is equivalent to $e_p^{APO}$=−3.7 (kcal/mol).

3. $e_p$=−7.4 (kcal/mol) for ATP, ADP, and ATP-γS. Nucleotide affinity for the group-2 pockets should be independent of the arginine fingers which allows us to determine $K_d(2)$ value in *Figure 3A* from a published result. In a previous study on the PAN complex which is the proteasomal ATPase homolog in archaebacteria and has a similar staircase architecture as the proteasome, the nucleotide off-rate was measured, and one time component (~3 s) was not affected by the arginine finger mutation on the PAN complex, which should be associated with the group-2 pockets (*Majumder et al., 2019*; *Kim et al., 2015*). The *APO* pocket parameters are also arginine-finger independent; however, its off-rate is about 100× higher, easily separable from this result. Therefore, this analysis identifies $K_d(2)$=3.3 µM, which gives $e_p$=−7.4 (kcal/mol).

4. $e_{Br,ADP}$=−0.52 (kcal/mol) $e_{Br,ATP}$=−2.1 (kcal/mol). From the single-molecule nucleotide proteasome interaction measurement, we obtained $e_{Br,ATP}-e_{Br,ADP}$=−1.58 kcal/mol. In a study using rabbit 26S proteasome, ATP-γS promotes the peptidase activity of proteasome likely by restructuring the ATPase complex (*Smith et al., 2011*). The EC$_{50}$ value of ATP-γS in this assay should be mapped to the strong pockets or group-1 in *Figure 3A*, which gives $e_{Br,ATP}+e_p$=−9.5 (kcal/mol) → $e_{Br,ADP}$=−0.52 (kcal/mol) & $e_{Br,ATP}$=−2.1 (kcal/mol). ATP-γS is assumed to have the same energy parameters as ATP.

5. eb=0 for all the interfaces. The basal energy represents in the nucleotide-independent affinity between ATPases. The RP tends to dissociate from the CP in the absence of nucleotide (*Hoffman and Rechsteiner, 1997*). The nucleotide-free RP structure suggests an open ring with broad ATPase interfaces, indicating the intrinsic affinity between ATPases is weak (~0 kT) (*Lu et al., 2017*). When simulating the transitions, we assigned a small random value (~0.2 kcal/mol) for each interface to resolve the ambiguity due to energy level degeneracy. We randomized these values in each simulation.

6. $k_{off}$ was determined for individual pocket according to the following equation.

$$k_{off} = k_{on} * e^{(E_p+E_{Br})/kT} * C_0 \tag{2}$$

where $E_p$, $E_{Br}$ are the pocket and bridge part of the free energy at a specific pocket. $C_0$ is the standard concentration (1 M).

## Determining the parameters related with substrate translocation

The transition rate between conformation A and B is generally described by the following Arrhenius equation:

$$r_{A-} = k_0 e^{(E_A - E_B - f_{AB} * d_{AB})/2kT} \tag{3}$$

where $E_A$, $E_B$ are the standard free energy of conformation A and B at a given nucleotide status; $f_{AB}$ is the average force on substrate during the translocation; We assigned two separated values of $f_{AB}$ for the forward and backward translocations to reflect the dissipative and conservative parts of this force; $d_{AB}$ is the translocation distance associated with the A→B conformational change (Materials and methods 'Defining a discrete conformational space of the proteasome ATPase complex by extrapolating the cryo-EM observations'). $k_0$ is a constant pertaining to the activation energy barrier. The factor '2' comes from detailed balance.

We determined the three parameters necessary for specifying the translocation kinetics: the average force exerted on the substrate peptide during forward (toward the CP) translocation; the average force during the backward translocation; the energy-barrier parameter $k_0$. We varied the concentration of ADP-Mg$^{2+}$ in the presence of 0.5 mM ATP, and measured the degradation rate or the equivalent translocation rate of ubiquitylated cyclinB-iRFP by purified human 26S proteasome as described in *Figure 3F*. We scanned the relevant range of the three parameters and determined the translocation rate ~ [ADP] at each parameter combination (Materials and methods 'Monte Carlo simulation of the FEL model of proteasome'). The best parameter trio matching the experimental observation at all ADP concentrations was chosen: 7 (index following Materials and methods 'Determing the FEL and kinetic parameters'). $f_{AB}$ during forward translocation=1.8 pN.

1. $f_{AB}$ during backward translocation=0.56 pN
2. $k_0$=72 s$^{-1}$
3. $k_h$=1.4 ATP/s/subunit, zero for ADP or ATP-γS. $k_h$ is the ATP hydrolysis rate of each pocket. The total rate of ATP hydrolysis was calculated in a simulation of translocation as described in Materials and methods 'Monte Carlo simulation of the FEL model of proteasome'. The hydrolysis rate was chosen to match the experimental values of 3.8–6 ATP/s/19S (*Peth et al., 2013b*). From current cryo-EM structures, we did not identify any evidence supporting allosteric effect to upregulate or downregulate the ATP hydrolysis activity at certain nucleotide pockets in the ATPase staircase. We therefore assigned the same hydrolysis rate to all the pockets at closed interfaces, since Arginine fingers are required for ATP hydrolysis (*Ogura and Wilkinson, 2001*; *Smith et al., 2011*; *Kim et al., 2015*). We have tested alternative assumptions by restricting the hydrolysis activity to the subunits close to the 20S and found that the qualitative features of the simulation results remained. Nonetheless, the current $k_h$ assignment provided the best overall quantitative consistency with the experimental data.

## Monte Carlo simulation of the FEL model of proteasome

The starting condition was set to the $E_{D1}$ state. The conformation of the ATPase hexamer evolved according to the FEL at the current nucleotide status and *Equation 2*. The nucleotide statuses at the six pockets independently evolved according to the kinetic constants at each pocket. We did not impose additional constraints or allosteric coupling between nucleotide pockets or with ATPase conformations.

For each simulation, the system was evolved for 400,000 steps corresponding to ~1000 s in physical time that is sufficient to achieve a steady-state distribution. A block diagram for the simulation algorithm was included in *Figure 2—figure supplement 2*.

For simulating the backward process, the forward translocation was set up zero when the translocated AAs≥$L_0$ which is the length of the peptide track. $L_0$ ranges from 20 to 30 AAs because of the uncertainty of defining the distance between PL1s to the proteolytic sites in the CP. Therefore, when the translocated AAs=0, we assume the substrate will escape from the proteasome. Substrate reentry is not considered due to a loss of the ubiquitin signal from the cyclinB segment.

For simulating a Walker-B motif mutant, the hydrolysis rate constant of the mutated subunit was set to zero.

For simulating Lid-ATPases interaction, the standard free energy of an $E_D$-like conformation that mimics the ATPase architecture in the $E_A$ states was lowered by an arbitrary value. In this conformation,

Rpt3's PL1 occupies the top position in the ATPase staircase while Rpt2 at the bottom and Rpt6 disengaged. We scanned this value from 1.6 to 4.0 kcal/mol for testing the consistency in each task.

## Parameter sensitivity analysis

The values of the parameters are either decreased or increased by 30%, except for $k_{on}$ which is varied by 3× due to the large uncertainty in its estimation. The translocation rates at all the nucleotide conditions in *Figure 4* are calculated using the FEL model and are compared with those at the original parameter values. The mean deviation of each parameter perturbation over the original translocation rate is calculated as the sensitivity score.

All the simulation was performed on the O2 cluster at Harvard Medical School using 48 cores, programmed in Matlab 2018 with the parallel computing toolbox. The source code is available from GitHub (*Ying, 2022*) or by request. Network figures were generated using Pajek (http://vlado.fmf. uni-lj.si/pub/networks/pajek/).

## Deriving a formula for the EC$_{50}$ value in the ATP titration experiment

The sequences of fundamental steps underlying a productive translocation were shown in *Figure 5—figure supplement 4*. The total time for one translocation step is

$$t_{tr} = 1/k_h^{all} + <1/r_{A-}> + 1/\left(k_{on}\left[ATP\right]\right) + 1/k_{off}^{APO} \tag{4}$$

where $k_h^{all}$ is the overall rate of ATP hydrolysis of the hexamer; ' $<1/r_{A \to B}>$ ' is the average conformational transition time.

Under limiting ATP concentration, $1/\left(k_{on}\left[ATP\right]\right)$ becomes significant. Therefore

$$EC_{50}\left(ATP\right) = \frac{1}{k_{on}\left(1/k_{off}^{APO} + 1/k_h^{all} + <1/r_{A-}>\right)} \sim \frac{k_h^{all}}{k_{on}} \tag{5}$$

## The translocation rate at different ATP concentrations as predicted by a strict sequential-transition model

The previously proposed sequential model does not contain quantitative details (*Dong et al., 2019*; *de la Peña et al., 2018*). As a comparison with the FEL model, we adopt a simple form of sequential model without losing generality. This sequential model involves six equivalent kinetic segments in a cycle; each contains two processes: 1. binding/unbinding of ATP; 2. ATP hydrolysis and ADP release.

If assuming functional symmetry of the six ATPases, the steady-state occupancy of $A_n = A_{n+1} \ldots = A$ and $B_n = B_{n+1} \ldots = B$, therefore under the steady-state condition:

$$0 = A * \left[ATP\right] * k_{on} - B * k_{off} - B * k_h \tag{6}$$

Considering A+B=C$^*$ is a constant due to the symmetry.

The rate of translocation B*k$_h$=$\frac{\left[ATP\right]*C^**k_{on}*k_h}{k_{off}+k_h+k_{on}*\left[ATP\right]}$ is a monotonic function of the ATP concentration.

## Interpreting the degradation kinetics of unfolding-limited substrates

The total turnover time of a cyclinB-DHFR-iRFP molecule under a substrate-saturating condition involves the time for translocation and the time for unfolding.

$$t_{total} = t_{tr}\left(\left[ATP\right]\right) + t_{uf}\left(\left[ATP\right]\right) \tag{7}$$

Unfolding of the DHFR domain mostly happens when DHFR is transiently unliganded. Because if that is the case, the inverse of the degradation rate of DHFR should linearly depend on the concentration of folic acid [FA] at any ATP concentration, according to *Equation 6*. This is indeed supported by the observation described in *Figure 5—figure supplement 1*.

$$t_{total} = 1/r_{deg} = 1/r_{tr}\left(\left[ATP\right]\right) + 1/r_{uf}\left(1 + \frac{k_{on}^{FA} * \left[FA\right]}{k_{off}^{FA}}\right) \tag{8}$$

where $r_{deg}$ is the degradation rate, $r_{uf}$ is the unfolding rate of unliganded DHFR on the ATPase complex. $k_{on}^{FA}$, $k_{off}^{FA}$ are the rate constants of folic acid on DHFR.

Substrate unfolding is likely achieved by the ATPase power strokes (*Iosefson et al., 2015*). Although the exact mechanism is still poorly understood, the unfolding rate should be generally in proportion to the ATPase activity in the absence of unfolding intermediates (*Martin et al., 2008*). The overall degradation rate is predicted to be

$$r_{deg} = \frac{r_{tr}\left([ATP]\right)}{1 + \frac{r_{tr}\left([ATP]\right)}{C_{uf}*r_h\left([ATP]\right)}\left(1 + \frac{k_{on}^{FA}*[FA]}{k_{off}^{FA}}\right)} \tag{9}$$

One extra parameter $C_{uf}$ was introduced as the unfolding rate coefficient in front of the ATPase activity $r_h$, which should generally depend on the folic acid concentration.

Translocation rate $r_{tr}$ and ATPase activity $r_h$ as functions of ATP concentrations were simulated using the FEL model. We adjusted $C_{uf}$ to match the measured degradation rate of cyclinB-DHFR-iRFP in the presence of folic acid (*Figure 4E*). The qualitative features of the degradation-rate curve in *Figure 4F, i*, that is, nearly-identical with that of the FA-free curve in the up-phase and weaker effect of high-ATP inhibition, are independent of the choice of $C_{uf}$.

### Measuring the ATPase activity of proteasome

100 nM purified human 26S proteasome was incubated in a buffer (1× UBAB, 0.5 mg/ml BSA, 0.05% NP-40, and 0.5 mM DTT) with varying concentrations of ATP-Mg$^{2+}$ for 30 min at 30°. No-proteasome controls were incubated under an identical condition. 20 μM denatured ovalbumin was added as a generic substrate of proteasome (*Cascio et al., 2001*). 30 nM and 50 nM proteasome was used for 100 μM and 300 μM ATP, respectively. After incubation, the concentration of free phosphate was quantified using Malachite green assay on a plate reader (BioTek Synergy H1).

### Measuring the residence time of an unfolding-resistant substrate on the proteasome

An iRFP substrate degradation assay is set up as described (Materials and methods 'iRFP substrate degradation assay'), involving 100 nM polyubiquitylated cyclinB-iRFP reporter substrate and 50 nM polyubiquitylated cyclinB-DHFR-iRFP$^{dark}$ as the competitor, either in the presence or absence of 300 nM methotrexate (MTX). cyclinB-DHFR-iRFP$^{dark}$ is nonfluorescent due to the lack of biliverdin in iRFP. The residence time $T_B$, that is, the mean time between when translocation reaches $L_0$ and escaping from the proteasome, of cyclinB-DHFR(MTX)-iRFP is calculated based on the formula:

$$T_B = \frac{\left(\frac{r_{RI0}}{r_{RI}} - 1\right)[R]}{[C]} T_R + \frac{r_{RI0}}{r_{RI}} \times T_{I0} - \frac{L}{v_t} \tag{10}$$

$r$RI0: reporter's degradation rate in the absence of MTX
$r$RI: reporter's degradation rate in the presence of MTX
$[R],[C]$: concentrations of reporter and competitor
$T$R: turnover time of the reporter (55 s)
$T$I0: turnover time of competitor in the absence of MTX (96 s)
$L$IN: length of the cyclinB: 90 AA $v_t$ : translocation rate (10.5 AA/s).

## Acknowledgements

The authors thank L Huang for sharing constructs for proteasome purification and thank B Schulman and N Brown for sharing constructs for APC/C purification. The authors are grateful for the critical reading and comments by M Kirschner, Y Tu, T Mitchison, D Finley, A Goldberg, L Bai, R Ward, J Yan, and P Ho. Portions of this research were conducted on the O2 High Performance Compute Cluster, supported by the Research Computing Group, at Harvard Medical School. This work is supported by an NIH R01 Grant to Dr. Lu (GM134064-01), an Edward Mallinckrodt, Jr foundation award to Dr. Lu, and a Harvard Medical School Dean's Initiative award to Dr. Finley and Dr. Lu.

## Additional information

### Funding

| Funder | Grant reference number | Author |
|---|---|---|
| National Institute of General Medical Sciences | GM134064-01 | Ying Lu |
| Edward Mallinckrodt, Jr. Foundation | | Ying Lu |
| Harvard Medical School | Dean's initiative award | Ying Lu |

The funders had no role in study design, data collection and interpretation, or the decision to submit the work for publication.

### Author contributions

Rui Fang, Conceptualization, Data curation, Formal analysis, Investigation, Software, Validation, Visualization, Writing - original draft, Writing - review and editing; Jason Hon, Data curation, Formal analysis, Investigation, Visualization, Writing - original draft, Writing - review and editing; Mengying Zhou, Data curation, Methodology, Validation, Visualization, Writing - original draft; Ying Lu, Conceptualization, Data curation, Formal analysis, Funding acquisition, Investigation, Methodology, Project administration, Resources, Software, Supervision, Validation, Visualization, Writing - original draft, Writing - review and editing

### Author ORCIDs

Ying Lu (ID) http://orcid.org/0000-0003-3516-7735

### Decision letter and Author response

Decision letter https://doi.org/10.7554/eLife.71911.sa1
Author response https://doi.org/10.7554/eLife.71911.sa2

## Additional files

### Supplementary files

• Supplementary file 1. ATPase conformations and steady-state occupancies. "Close interface": the six digits indicate the Rpt6-Rpt3, Rpt3-Rpt4, Rpt4-Rpt5, Rpt5-Rpt1, Rpt1-Rpt2, Rpt2-Rpt6 interfaces. "0": open; "1": closed. "Engaged ATPases": the six digits indicate Rpt6, Rpt3, Rpt4, Rpt5, Rpt1, Rpt2. "0": disengaged subunit; "1": engaged subunit. "PL1 registry": the six digits indicate the PL1s on Rpt6, Rpt3, Rpt4, Rpt5, Rpt1, Rpt2. "1~5": part of the staircase architecture. "1" is closest to the CP; and "5" is farthest from the CP. "7": disengaged PL1 at the top registry. "Steady-state occupancy (%)": the steady-state occupancy of each conformation in a FEL simulation. 2. Steady-state transition rates among ATPase conformations in the FEL model. "Total transition": the total rate of transitions from conformation 1 to conformation two and reverse. Numbers are normalized by the highest value set to 100. "Net transition": the absolute value of the rate difference between conf1-conf2 and conf2-conf1 transitions. Numbers are normalized by the same factor as above. "Translocation": F: the net effect of conf1-conf2 transitions is a forward translocation of substrate; "B": the net effect is a backward translocation; "N": this conformational transition does not lead to translocation.

• Transparent reporting form

### Data availability

All data generated or analyzed during this study are included in the manuscript and supporting files. Source data files have been provided for Figures 3,4,5. The original images for the single-molecule experiments are available on Dryad (https://doi.org/10.5061/dryad.t1g1jwt2t). The source code is available in GitHub (https://github.com/luyinghms/Proteasome-FEL-model.git, copy archived at swh:1:rev:b0bda002256c37d595376cb77b39f0f40ca02fe5).

The following dataset was generated:

| Author(s) | Year | Dataset title | Dataset URL | Database and Identifier |
|---|---|---|---|---|
| Lu Y | 2021 | Single molecule images for: An empirical energy landscape reveals mechanism of proteasome in polypeptide translocation | https://doi.org/10.5061/dryad.t1g1jwt2t | Dryad Digital Repository, 10.5061/dryad.t1g1jwt2t |

The following previously published dataset was used:

| Author(s) | Year | Dataset title | Dataset URL | Database and Identifier |
|---|---|---|---|---|
| Dong Y, Zhang S, Wu Z, li X, Wang W, Zhu Y, Stoilova-McPhie S, lu Y, Finley D, Mao Y | 2019 | cryo-structures of substrate-engaged human 26s proteasome | https://www.rcsb.org/6MSE | RCSB Protein Data Bank, 6MSE |

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
