## [Editor Report]

The present work is important for using innovative computational approaches and biochemical analyses to help to explain how hexameric peptide translocases and unfoldases belonging to AAA+ ATPases couple nucleotide turnover to directed chain movement. The work sheds light on understanding not only normal, processive translocation but also how the motors can operate with a defective subunit.

---

## [Decision Letter]

**Decision letter after peer review:**

Thank you for submitting your article "An Empirical Energy Landscape Reveals Mechanism of Proteasome in Polypeptide Translocation" for consideration by *eLife*. Your article has been reviewed by 3 peer reviewers, one of whom is a member of our Board of Reviewing Editors, and the evaluation has been overseen by David Ron as the Senior Editor. The following individual involved in review of your submission has agreed to reveal their identity: Gregory R Bowman (Reviewer #3).

Essential revisions:

All three referees felt that the paper is extremely technical and difficult to follow for those who are not computationalists or steeped in the minutia of AAA+ unfoldases. They also noted that it is unclear how the findings advance the conceptual or mechanistic understanding of AAA+ unfoldases beyond what is currently accepted in the field. These criticisms were counter-balanced by a sense that there is novelty and impact in the findings that have the potential for broad appeal. In addition to the specific comments noted below, the general writing and organization of the manuscript need to be substantially improve to more clearly describe the computational approach and the data analysis, and to better frame the explanatory power, impact and novelty of the results.

*Reviewer #1 (Recommendations for the authors):*

Page 6 and methods. Is it legitimate to exclude the chemical energy of ATP from the calculations?

P. 7. Concerning the statement "…we made the simplifying assumption that all six ATPases share an identical set of parameters. This assumption does not contradict the experimental finding of functional disparity among the six ATPases". What is the evidence for this claim?

Many parts of section II are dense with jargon and difficult to follow for the non-specialist. The language of this section should be simplified and clarified. Blanket statements and assumptions need to be accompanied with supporting evidence and arguments.

P. 7 and Figure 3B. There is ample evidence of cooperativity between nucleotide and substrate binding throughout AAA+ ATPases; i.e., the presence of peptide can alter the affinity for ATP and ADP and vice versa. However, it appears that the measurement of the nucleotide binding affinities was carried out using substrate-free (no peptide) proteosomes. How can the authors know that the values they obtained are relevant to when the ATPases are in a translocation state? And if these values are critical for estimating bridge energies, how can one be sure that these estimations have not thus been miscalculated?

Figure 2C. What is to be made of the observation that the TDETTT configuration leads to extremely low free energies only for states 3 and 8? This outcome could be taken to imply that the FEL analysis is being biased by the starting models; i.e., because the models were based on ED2, which is in state3 and not far off from state 8. Is there a counterargument for this concern?

Figure 3. Many of the labels, particularly in panels B and C, are too small to be legible.

*Reviewer #2 (Recommendations for the authors):*

I'm impressed by the ambition of the manuscript – to provide a full quantitative mechanistic model of proteasomal protein degradation. The experimental approach is very sophisticated and overall well thought through. However, I was unable to assess the quality of the data and reliability of the conclusions because I felt that the interpretation of the results and was not given enough room. In addition, I am not convinced that the conclusions discussed provide novel insights into the mechanism of the proteasome.

*Reviewer #3 (Recommendations for the authors):*

I expect other high energy states are present (and may even be important) but have low enough probability that this model still performs well. For example, the authors ignore states with more than 3 open interfaces. Presumably they are present, but I expect at much lower probabilities. Could the authors comment on how much lower probability these conformations would have than those included in the model, and how reasonable ignoring them is on that basis? It would also be useful to acknowledge that there may be other high-energy intermediates (e.g. as the PL1 loops move) that would be important to consider to explain the effects of specific mutations.

Assuming all the subunits have identical behavior does remarkably well. I'm curious how much asymmetry one could introduce without qualitatively changing the model (e.g. keeping the dominant states and transitions). I think an exhaustive study is beyond the scope of this work, but I would be curious to see a little bit of data. For example, would the depiction in Figure 6 look more or less the same if each interface had a different energetic separation between the open/closed states? E.g. I could imagine having a different e_b parameter for each interface, where each is chosen by multiplying the constant used so far by a random factor between 0.5 and 2. It should be easy to do this many times. Ideally the authors could make a statistical statement on how sensitive the topology is across this distribution of models, but even showing some examples to give a qualitative sense of the variability in an SI figure would probably be sufficient for this first paper.

---

## [Author Response]

Essential revisions:All three referees felt that the paper is extremely technical and difficult to follow for those who are not computationalists or steeped in the minutia of AAA+ unfoldases. They also noted that it is unclear how the findings advance the conceptual or mechanistic understanding of AAA+ unfoldases beyond what is currently accepted in the field. These criticisms were counter-balanced by a sense that there is novelty and impact in the findings that have the potential for broad appeal. In addition to the specific comments noted below, the general writing and organization of the manuscript need to be substantially improve to more clearly describe the computational approach and the data analysis, and to better frame the explanatory power, impact and novelty of the results.

This interdisciplinary study integrates concepts and approaches in multiple fields and introduced novel computational and experimental methods. This may have created difficulties in understanding our work. While some technical terms may be necessary, we replaced them with nontechnical language as far as possible without compromising accuracy. In this revision, we:

– Substantially rewrote the introduction to include a non-technical summary of the work.

– Updated Figure 4-Figure supp. 1 (previously S7) to include the workflow of the entire study, including the key observations and measurements for model construction, the experimental validations of the predictions and the major insights into the mechanism of proteasomal ATPases. This should improve the communication of the logic and the structure of this work.

– Improved the structure of the writing, especially in section II. We use the starting sentence of each paragraph to summarize the main results and conclusions, so that readers can understand the main thrust of the discussion without reading every detail.

– Substantially rewrote the Discussion to better frame the novelty and contribution of this study. Major biological insights are emphasized and discussed. We also better discussed the caveats of this method.

– Attempted to clarify points that caused confusion.

– Updated Figure 2, Figure 3, Figure 6 and Figure 2-Figure supp. 2 (previously S4) according to the suggestions.

A major contribution of our study is demonstrating the feasibility of performing a meaningful simulation of the dynamics of a large ATPase system in a biologically-relevant time scale. Therefore, most of the manuscript is dedicated to introducing the FEL-based simulation approach and validating the predictions. Due to limitations in the allowed length of the manuscript, we are only able to elaborate on the most important biological insights that have not been identified previously in the proteasome system. The model and the simulated dynamics carry insights that are not discussed in the manuscript. One example is the results from the analysis suggested by reviewer #3. We will make the computational code available and easy to execute, so that readers can explore the full potential of the model.

Reviewer #1 (Recommendations for the authors):Page 6 and methods. Is it legitimate to exclude the chemical energy of ATP from the calculations?

The FEL model does not force any coupling between nucleotide and conformation, so that the nucleotide status is stable during an instantaneous change of the ATPase conformation. The rate of a conformational transition depends only on the free energy difference but not on the absolute free energy values (See Methods). Therefore, whether we include the chemical-energy term or not does not affect the simulation results. Nucleotide cycles are described by rate equations, which also do not require a chemical-energy term.

P. 7. Concerning the statement "…we made the simplifying assumption that all six ATPases share an identical set of parameters. This assumption does not contradict the experimental finding of functional disparity among the six ATPases". What is the evidence for this claim?

We thank the reviewer for pointing out this awkwardly-worded sentence. We reworded this part to clarify. Essentially, we find that a model using the same parameters for all 6 ATPases can nevertheless recapitulate the degradation kinetics, the asymmetric cryo-EM occupancies and effects of Walker-B mutations. This approximation is also technically important to avoid overinterpretation, because incorporating Lid-ATP interaction requires only one more parameter and in contrast 18 more parameters are required to describe the ATPases’ differences. We therefore believe that this approximation is useful. Please note that we are not saying that the parameters are *actually* identical for the six ATPases (which of course is unlikely). All we are saying is that any parameter difference between the ATPases turns out to be unimportant in this context. We are aware that there are published results reporting functional asymmetry of these ATPases in the presence of other mutations e.g. Walker-A or pore loops. Due to the known proteasome assembly defects of these mutants or ambiguity of mapping a mutation to the model, we are unable to clearly evaluate the model in these contexts. We discuss this question in the manuscript.

Many parts of section II are dense with jargon and difficult to follow for the non-specialist. The language of this section should be simplified and clarified. Blanket statements and assumptions need to be accompanied with supporting evidence and arguments.

We hope that the writing in section II is now substantially improved. We specifically clarified the basis for each statement by citing a published result or an observation in this study.

P. 7 and Figure 3B. There is ample evidence of cooperativity between nucleotide and substrate binding throughout AAA+ ATPases; i.e., the presence of peptide can alter the affinity for ATP and ADP and vice versa. However, it appears that the measurement of the nucleotide binding affinities was carried out using substrate-free (no peptide) proteosomes. How can the authors know that the values they obtained are relevant to when the ATPases are in a translocation state? And if these values are critical for estimating bridge energies, how can one be sure that these estimations have not thus been miscalculated?

The general consistency between the model predictions and experimental results (degradation kinetic and etc. ) supports the relevance of these affinity constants. The model has nine parameters, three of which are directly related to these affinity constants. Significant deviation from the actual values is unlikely to yield predictions that are consistent with the experiments (Figure 5-Figure supp. 5). While it is possible that these three parameters change when the proteasome moves into the translocation state, we nevertheless observe good agreement between model predictions and experimental observations. This indicates that the parameters we derived from observations of the substrate-free proteasome are close approximations to the true values. Undoubtedly, the model will be improved in the future by the incorporation of more specific measurements of affinity constants of each pocket.

Structural comparisons show that there is no significant change of the nucleotide pocket geometry between the resting and translocating states, indicating that the current affinity measurement should be a good approximation of the actual values. To test this, we analyzed the nucleotide-pocket geometries in the cryo-EM structures either with or without substrate peptide in the ATPase channel, and compared these geometries in Figure 2- figure supp. 2. While nucleotide-bound and unbound pockets are clearly different, we do not observe any significant change caused by substrate among pockets with the same nucleotide status.

Please note that the current model does not necessarily contradict the general observation that substrate may alter nucleotide affinity, because a higher rate of nucleotide turnover can be caused by accelerated ATPases dynamics in the presence of substrate. The current affinity measurement is an equilibrium method requiring ADP or ATP-γS and no substrate, which greatly simplifies data interpretation. How substrate leads to accelerated ATPase dynamics is unclear and is beyond the scope of the current study. In our model, efficient nucleotide turnover requires disengaged ATPases that have low affinity for nucleotides. We speculated in Discussion that the disengaged ATPases in the translocating states may have lower nucleotide affinity than the disengaged ones in the resting states, which may lead to a higher rate of nucleotide turnover in the presence of substrate.

Figure 2C. What is to be made of the observation that the TDETTT configuration leads to extremely low free energies only for states 3 and 8? This outcome could be taken to imply that the FEL analysis is being biased by the starting models; i.e., because the models were based on ED2, which is in state3 and not far off from state 8. Is there a counterargument for this concern?

As mentioned in the manuscript, the model does not force any coupling between nucleotide and conformation, and therefore, is unbiased in this sense. The concern may come from the fact we use structural information to build the model. As summarized in Figure 4-Figure supp. 1, we only extracted a few simple rules from these cryo-EM structures, without referring to any particular state. Any single state, such as ED1 or ED2, in the cryo-EM dataset can be omitted from the analysis and essentially the same model and predictions would result.

The energy landscape we give in Figure 2C is only an example: we could have used any nucleotide state in this example. The reason we chose the same nucleotide state as in the ED1 state was to allow comparison with structural observations. Although all possible combinations of nucleotide and conformation are allowed in the model, the fact that the energy-minimum ATPase conformation at the nucleotide status “TDETTT” is identical to the ED1 conformation in fact supports the FEL model, and may explain the coincidence of this ATPase conformation and the “TDETTT” status in the previous structural study.

Figure 3. Many of the labels, particularly in panels B and C, are too small to be legible.

We increased the font in Figure 3.

Reviewer #2 (Recommendations for the authors):I'm impressed by the ambition of the manuscript – to provide a full quantitative mechanistic model of proteasomal protein degradation. The experimental approach is very sophisticated and overall well thought through. However, I was unable to assess the quality of the data and reliability of the conclusions because I felt that the interpretation of the results and was not given enough room. In addition, I am not convinced that the conclusions discussed provide novel insights into the mechanism of the proteasome.

We realize that we have attempted to provide a large amount of information in a single paper. Without describing the FEL model in detail, the data interpretation and mechanistic insights are not possible to assess. Therefore, we devoted a significant part of the manuscript to establishing and validating the FEL model, leaving less room for biological implications than would be ideal. We have done our best to simplify and streamline the presentation in this revised manuscript, as described at the beginning of this letter.

In this paper we show that the FEL model provides a clear and unified mechanistic interpretation that is compatible with experimental observations and rooted in fundamental physical/chemical processes. Certainly, the model has caveats that may lead to alternative interpretations of some results. In our revision we substantially expanded the discussion about the limitations of this model and offered alternative interpretations of the results affected by the limitations. We list the main biological insights that may interest proteasome biologists:

– A transition sequence of the ATPase conformation during substrate translocation (Figure 6A, B), which is compatible with both the cryo-EM structures and the kinetic results in this and previous studies. This transition sequence is different from the previous sequential model, the only previous model (to our knowledge) that has been proposed for the proteasome.

– An explanation for the cooperative mechanism of the ATPase subunits. The FEL model suggests that the proteasome has a unique energy-minimum conformation for each typical nucleotide-bound status. This uniqueness leads to high ATPase cooperativity because it entails define transitions of the ATPases’ conformation at the change of nucleotide status. This is the first plausible explanation of the cooperativity between these ATPases, to our knowledge.

– An explanation for the differences between these ATPases in functional studies. Our study suggests that the Lid-ATPase interaction alters the ATPase dynamics, giving rise to predictions that are consistent with conformational occupancies of both WT proteasome and the Walker-B mutants and are also qualitatively consistent with the growth phenotypes due to Walker-B mutations. Previous interpretations often invoke intrinsic differences between the six ATPases. However, the contribution or phenotype of each ATPase often varies significantly in different assays, leading to no consistent interpretation of the symmetry-breaking mechanism. Please note that this result does not dismiss the possibility that there are in fact intrinsic differences among these ATPases – we simply argue that such differences are not required to explain the phenotypes observed. Future studies may identify phenotypes that can only be explained by incorporating the intrinsic differences among the ATPases in the FEL model.

Reviewer #3 (Recommendations for the authors):I expect other high energy states are present (and may even be important) but have low enough probability that this model still performs well. For example, the authors ignore states with more than 3 open interfaces. Presumably they are present, but I expect at much lower probabilities. Could the authors comment on how much lower probability these conformations would have than those included in the model, and how reasonable ignoring them is on that basis?

In the original work, we constrained the total number of open interfaces in each conformation to be either 2 or 3 and also excluded five conformations with 3+3 or 2+2+2 symmetry; let us call this model model 0. We removed all these constraints in a modified model designated model 1**.** We went through the same process of reevaluating the three translocation-related parameters in model 1. As the reviewer suggested, allowing these high-energy states does not significantly change the steady-state conformational distribution. The occupancy of these high-energy states is also very small, in total less than 5% (Author response image 1). The most populated conformation is the 3+3 (two 3xATPase staircases, separated by open interfaces), which represents 2.3% of the conformations in model 1.

**Author response image 1. sa2fig1:** Testing alternative FEL models. (A) Steady-state occupancies for different conformations. See text for the definition of each model. B-D. Comparison of different models in predicting translocation rates in various nucleotide conditions.

It would also be useful to acknowledge that there may be other high-energy intermediates (e.g. as the PL1 loops move) that would be important to consider to explain the effects of specific mutations.

We went through the data analysis pipeline described in the paper using model 1. Interestingly, it failed to quantitatively recapitulate the translocation measurements, although the trend remains. We surmise that this may be because the 3+3 conformation (2.3% of the total) leads to ambiguity in assigning which ATPase staircase should bind substrate. Next, we developed a new modified model excluding only the 3+3 conformation model 2. Model 2 recapitulates the experimental measurements similarly to model 0 (Author response image 1). We thank the reviewer for the insightful comments.

Assuming all the subunits have identical behavior does remarkably well. I'm curious how much asymmetry one could introduce without qualitatively changing the model (e.g. keeping the dominant states and transitions). I think an exhaustive study is beyond the scope of this work, but I would be curious to see a little bit of data. For example, would the depiction in Figure 6 look more or less the same if each interface had a different energetic separation between the open/closed states? E.g. I could imagine having a different e_b parameter for each interface, where each is chosen by multiplying the constant used so far by a random factor between 0.5 and 2. It should be easy to do this many times. Ideally the authors could make a statistical statement on how sensitive the topology is across this distribution of models, but even showing some examples to give a qualitative sense of the variability in an SI figure would probably be sufficient for this first paper.

We tested the effect of varying e_b. In the model, we let e_b(i) for the i^th^ ATPase = α×δ^i^_._ δ^i^ is a random number from 0 to 1 and α is a scaling constant. In one test, we varied α while fixing δ^i^ (Author response image 2A). In Author response image 2B, we tested different δ^i^ with the same α value. The result suggests that a small variation of e_b causes minor changes in the predicted dynamics, while large e_b variation may significantly affect the dynamics.

We also studied whether e_b variation also affects the ATPase dynamics when the Lid-ATPase interaction is included in the model (Author response image 2C). Interestingly, the Lid-ATPase interaction significantly blunts the effect of e_b variation even at a large amplitude of 1 kcal/mol (comparing Author response image 2C and Figure 6-Figure supp. 4). We include this result in Figure 6-Figure supp. 6.

**Author response image 2. sa2fig2:** Global dynamical space in the presence of variations in e_b. (A) Varying the amplitude α of e_b variation while fixing the ratios. In this example, δ^i^=(0.45,0.15,0.83,1.0, 0.68,0.88). B. Two cases with alternative δ^i^ and the same amplitude. C. The same e_b setting as in A(middle), but including the Lid-ATPase interaction as in Fig. 6-fig. supp. 4. Results are presented as in Fig. 6.